# *Let-7* microRNA-dependent control of leukotriene signaling regulates the transition of hematopoietic niche in mice

Xuan Jiang[1], John S. Hawkins[1], Jerry Lee[1], Carlos O. Lizama[1], Frank L. Bos[1], Joan P. Zape[1], Prajakta Ghatpande[1], Yongbo Peng[1], Justin Louie[1], Giorgio Lagna[1], Ann C. Zovein [1,2] & Akiko Hata [1]

Hematopoietic stem and progenitor cells arise from the vascular endothelium of the dorsal aorta and subsequently switch niche to the fetal liver through unknown mechanisms. Here we report that vascular endothelium-specific deletion of mouse *Drosha* (*Drosha*[cKO]), an enzyme essential for microRNA biogenesis, leads to anemia and death. A similar number of hematopoietic stem and progenitor cells emerge from Drosha-deficient and control vascular endothelium, but *Drosha*[cKO]-derived hematopoietic stem and progenitor cells accumulate in the dorsal aorta and fail to colonize the fetal liver. Depletion of the *let-7* family of microRNAs is a primary cause of this defect, as it leads to activation of leukotriene B4 signaling and induction of the α4β1 integrin cell adhesion complex in hematopoietic stem and progenitor cells. Inhibition of leukotriene B4 or integrin rescues maturation and migration of *Drosha*[cKO] hematopoietic stem and progenitor cells to the fetal liver, while it hampers hematopoiesis in wild-type animals. Our study uncovers a previously undefined role of innate leukotriene B4 signaling as a gatekeeper of the hematopoietic niche transition.

[1] Cardiovascular Research Institute, University of California, San Francisco, San Francisco, CA 94143, USA. [2] Department of Pediatrics, Division of Neonatology, University of California San Francisco School of Medicine, San Francisco, CA 94143, USA. Correspondence and requests for materials should be addressed to A.H. (email: akiko.hata@ucsf.edu)

Hematopoietic progenitors first arise in the yolk sac (YS), initially as primitive erythrocytes and megakaryocytes at embryonic day (E) 7.25, followed by transient definitive erythroid/myeloid progenitors at E8.25[1–3]. During a limited developmental window, definitive hematopoietic stem and progenitor cells (HSPCs) emerge as clusters from the dorsal aorta (DA) in the aorta/gonad/mesonephros (AGM) and in the vitelline and the umbilical arteries (VA + UA),[1–3]. A complex and finely tuned process choreographs the emergence of HSPCs from the vascular bed. First, specialized endothelial cells (ECs), designated hemogenic endothelium (HE), undergo an endothelial-to-hematopoietic transition (EHT) and generate HSPCs. HSPCs are assembled as clusters attached to the arterial lumen, and remain attached until an unknown signal instructs them to detach from the vascular bed, enter the circulation, and migrate to the fetal liver (FL)[4–8]. By E13.5 the FL becomes the dominant hematopoietic organ in the embryo[2]. Recent studies have elucidated the involvement of various signals, including innate inflammatory signals, such as interferons and tumor necrosis factorα, in the early steps of de novo HSPC production from the HE in mouse and zebrafish[9–12]. However, the molecular and biochemical pathways that control subsequent events, such as HSPC maturation, release into circulation, and migration to the FL, are poorly understood due to lack of reported mutant abnormalities at stages that follow HSPC emergence.

Small non-coding microRNAs (miRNAs) mediate post-transcriptional gene regulation and are indispensable for normal embryogenesis and maintenance of homeostasis in every organ[13–16]. Although a number of miRNAs have been shown to be critical for hematopoiesis[16], little is known about the role of miRNAs in early embryonic HSPC generation. miRNA biogenesis requires the essential enzyme Drosha for cleavage of primary miRNA transcripts (pri-miRNAs) and generation of precursor miRNAs (pre-miRNAs) in the nucleus[13, 17]. Thus, deletion of Drosha leads to the depletion of nearly the entire miRNA population, thus enabling an assessment of the function of global miRNA-dependent gene regulation[13]. In this study we generated mice in which Drosha is conditionally deleted (Drosha^cKO) in the vascular endothelium (VE). We report that Drosha^cKO mice die prematurely due to a complete failure of definitive hematopoiesis, despite normal specification of HE and emergence of HSPC clusters. We link this phenotype in part to the elevated production of leukotriene B4 (LTB4) in HSPC clusters leading to increased adhesion to the VE and failure to migrate to the FL. Our study demonstrates the innate LTB4 signaling as determinants of spatiotemporal control in the transition process of HSPCs from the AGM to the FL.

## Results

**Loss of endothelial drosha leads to hematopoietic defects.** To elucidate the physiological function of miRNAs in definitive hematopoiesis, a VE-specific Drosha knockout mouse (Drosha^fl/fl; Cdh5-Cre^+, hereafter referred to as cKO) was generated by crossing Drosha^fl/fl mice[18] to a Cdh5-Cre transgenic mouse line[19]. Cdh5-Cre drives the expression of Cre recombinase in ECs at E8,

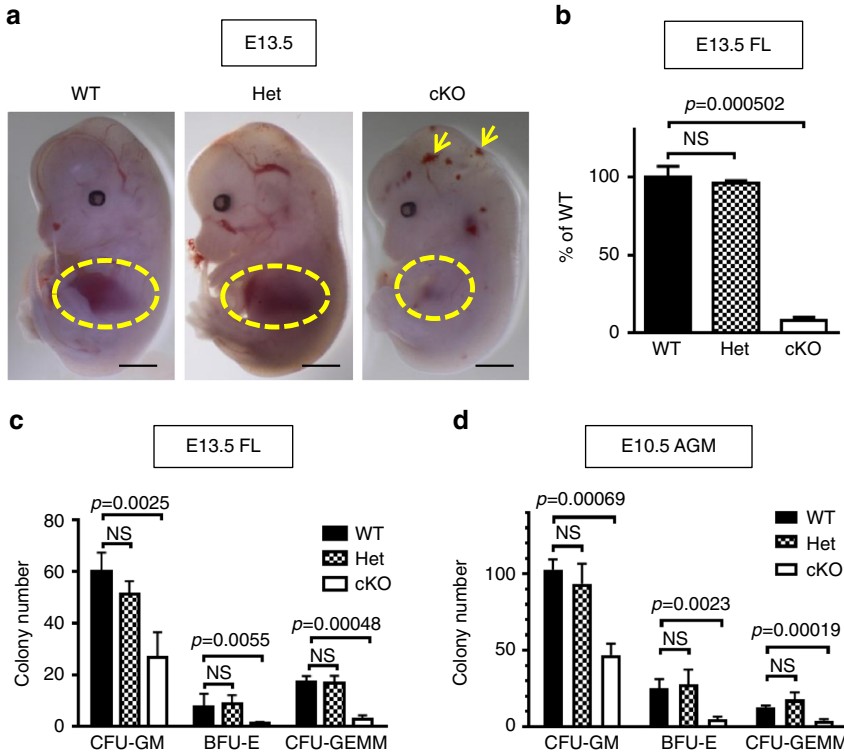

**Fig. 1** Hematopoietic defects in endothelial-specific deletion of Drosha. **a** Representative images of E13.5 WT, Het and cKO embryos indicate subcutaneous hemorrhage (*arrows*) and FL (*circles*) anemia in cKO embryos. Scale bars: 2 mm. **b** The cellularity of fetal liver from E13.5 WT, Het or cKO embryos was analyzed. Total number of DAPI⁻ live cells in the fetal liver was counted by flow cytometry. The number of cells in WT liver was set at 100% and the relative number in Het or cKO liver was shown as Mean ± SEM; p values were generated by unpaired Student's t test . NS, not significant. n = WT: 4 embryos, Het: 3 embryos, cKO: 4 embryos. 3 litters. **c** The fraction (0.13%) of cells from the FL of E13.5 WT (Drosha^fl/+ or Drosha^fl/fl), Het (Drosha^fl/+;Cdh5-Cre^+), or cKO (Drosha^fl/fl;Cdh5-Cre^+) embryos was subjected to CFU assay. The experiments were performed in duplicates. Colony counts of three progenitors (BFU-E, CFU-GM, and CFU-GEMM) were plotted as Mean ± SEM; p values were generated by unpaired Student's t test . n = WT: 8 embryos, Het: 4 embryos, cKO: 4 embryos. 2 litters. **d** 50% of cells from E10.5 AGM from WT, Het, or cKO embryos were subjected to CFU assay. Results of duplicate experiments were shown as Mean ± SEM; p values were generated by unpaired Student's t test . n = WT: 10 embryos, Het: 2 embryos, cKO: 4 embryos. 3 litters

thus Drosha is expected to be depleted from both embryonic and extraembryonic hemogenic tissues in cKO animals[19]. The EC-specific Cre recombinase activity was confirmed by using a Cre-reporter (*Rosa26*[mT/mG]) mouse in which GFP (*green*) is expressed upon Cre recombination, and tdTomato (*red*) is expressed in non-recombined tissues[20]. At E10.5, ECs along the DA in the AGM were GFP-positive, a sign of successful Cre excision (Supplementary Fig. 1a). Quantitative-RT-PCR (qRT-PCR) analysis confirmed an ~70% reduction of *Drosha* mRNA in EC populations (CD31[+]CD45[−]Kit[−]) in cKO compared to control

mice (hereafter referred to as Ctr), which include Het (*Drosha*[fl/+]; *Cdh5-Cre*[+]), and WT (*Drosha*[fl/+] or *Drosha*[fl/fl]) (Supplementary Fig. 1b). At E13.5, FL in cKO embryos appeared smaller and paler compared to Ctr (Fig. 1a and Supplementary Fig. 1c, *yellow circle*). The total number of liver cells in cKO was only 9% of Ctr (Fig. 1b), confirming the smaller cKO liver size (Fig. 1a and Supplementary Fig. 1c). Because the liver is the predominant site of erythroid development at E13.5[1–3], the liver phenotype in cKO suggested a hematopoietic defect. Therefore, we evaluated the number of Ter119[+] cells, which include mature erythroid cells

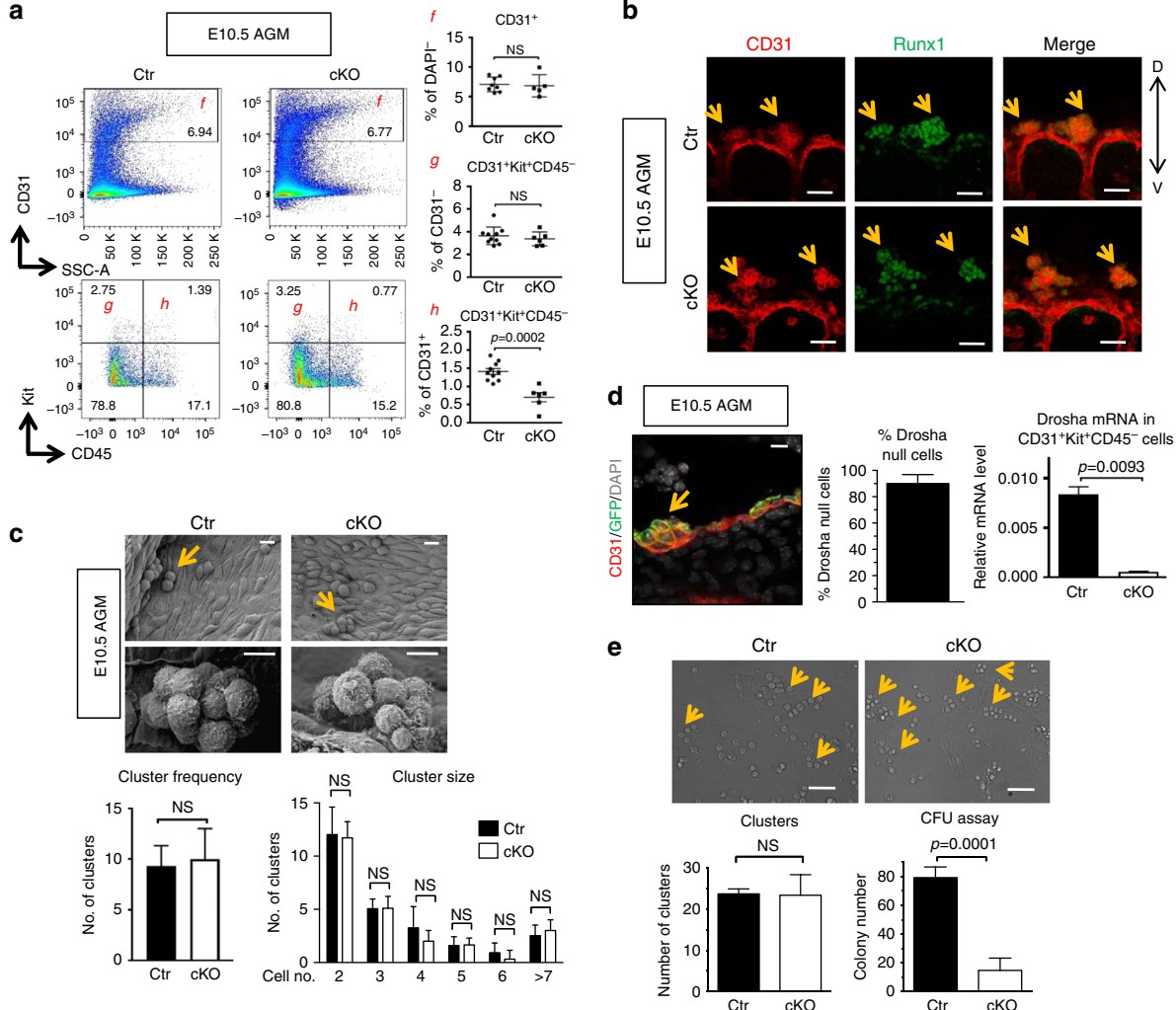

**Fig. 2** Normal emergence of intra-aortic HSPC clusters in Drosha cKO embryos. **a** Endothelial cells (CD31[+]), HSPC clusters (CD31[+]Kit[+]CD45[−]) and maturing HSPCs (CD31[+]Kit[+]CD45[+]) derived from AGMs of E10.5 Ctr or cKO embryos were quantitated by flow cytometry and shown as a frequency (%) of ECs among DAPI[−] cells, CD31[+]Kit[+]CD45[-] among CD31[+] (g), and CD31[+]Kit[+]CD45[+] among CD31[+] cells (h) (Mean ± SEM); *p* values were generated by unpaired Student's *t* test . *Scale bars*: 10 μm. n = Ctr: 10 embryos, cKO: 5 embryos. 3 litters. **b** Whole-mount IF staining AGMs from E10.5 Ctr or cKO embryos with anti-Runx1 (*green*) and anti-CD31 (*red*) antibodies. *Yellow arrows*: cluster cells. *D, V* indicate dorsoventral axis. *Scale bars*: 50 μm. **c** Representative scanning electron microscopy images of intra-aortic cluster cells in E10.5 AGM from Ctr or cKO embryos (*top*). Arrows: cluster cells. The cluster frequency panel (*bottom left*) indicates the number of cluster per 1000 ECs (Mean ± SEM). In the cluster size panel (*bottom right*), the number of cluster with different sizes per 1000 ECs was plotted (Mean ± SEM); *p* values were generated by unpaired Student's *t* test . Five images per embryo were taken and counted. *Scale bars*: 10 μm. n = Ctr: 5 embryos, cKO: 5 embryos. 3 litters. **d** A representative image of transverse section from E10.5 AGM of Drosha cKO with Cre recombinase reporter (GFP) allele (*Drosha*[fl/fl];*Cdh5-cre*[+];*Rosa26*[m/+], left panel) (*left*). Arrow: cluster cells (*left*). The fraction (%) of Cre active (GFP[+]) cells among ECs (CD31[+]) was plotted as Mean ± SEM (*middle*); *p* values were generated by unpaired Student's *t* test . *Right panel*: Six images were taken from each embryo. n = cKO: 2 embryos. 1 litter. qRT-PCR analysis of *Drosha* mRNAs relative to *GAPDH* mRNAs in HSPC clusters (CD31[+]Kit[+]CD45[−]) sorted from E11.5 Ctr or cKO embryos (Mean ± SEM); *p* values were generated by unpaired Student's *t* test . *Scale bars*: 10 μm. n = Ctr: 12 embryos. cKO: 6 embryos. 3 litters. **e** Endothelial (CD31[+]) cells sorted from E10.5 AGM from Ctr or cKO embryos were co-cultured with OP9-DL1 cells for 7 days. Representative images of cluster cells that emerged from ECs at day 5 are shown (*top*). Number of clusters per view under ×20 objective on day 7 were quantitated and shown as Mean ± SEM (*bottom left*). At day 7, all HSPC clusters from the co-culture were harvested and subjected to CFU assay. Total colony counts were presented as Mean ± SEM (*bottom right*); *p* values were generated by unpaired Student's *t* test . NS, not significant. *Scale bars*: 50 μm

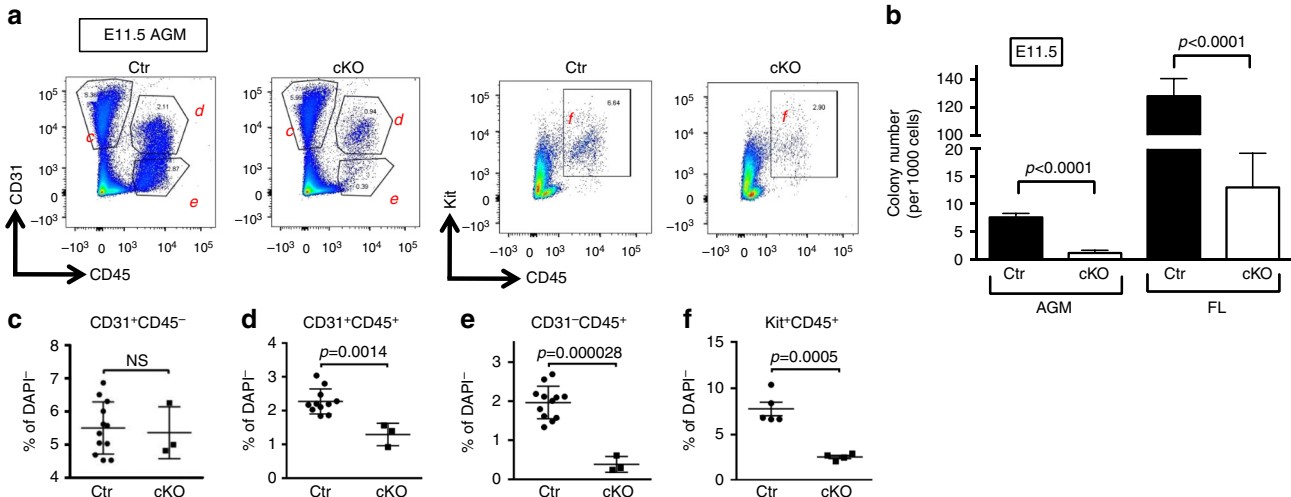

**Fig. 3** Impairment of HSPC maturation in Drosha cKO embryos. **a** Flow cytometric analysis of ECs (CD31$^+$CD45$^-$), HSPCs (CD31$^+$CD45$^+$ and Kit$^+$CD45$^+$) and mature hematopoietic population (CD31$^-$CD45$^+$) derived from E11.5 Ctr or cKO AGMs and shown as a frequency (%) each population among live cells (DAPI$^-$)(Mean ± SEM); $p$ values were generated by unpaired Student's $t$ test. **c–e** $n$ = Ctr: 12 embryos, cKO: 3 embryos. 2 litters. **f** $n$ = Ctr: 5 embryos, cKO: 4 embryos. 2 litters. NS, not significant. **b** Hematopoietic (CD45$^+$) cells were sorted from E11.5 AGM or FL, followed by CFU assay. Total counts of colonies of three progenitors (BFU-E, CFU-GM, and CFU-GEMM) were plotted as Mean ± SEM; $p$ values were generated by unpaired Student's $t$ test . For FL, $n$ = Ctr: 8 embryos, cKO: 5 embryos. 2 litters. For AGM, $n$ = Ctr: 11 embryos, cKO: 4 embryos. 2 litters

and erythroid precursors[21], in the FL by flow cytometry. The cKO livers contained only half the number of Ter119$^+$ cells as compared to controls (Supplementary Fig. 1d), confirming defective erythropoiesis in cKO, as is consistent with the pale coloration of the cKO livers (Fig. 1a and Supplementary Fig. 1c). To evaluate the functional ability of cKO livers, colony formation unit assays (CFU assays) were performed and compared to Ctr (Het or WT) at E13.5 (Fig. 1c). The number of colonies of three lineages (CFU-GM, BFU-E, and CFU-GEMM) from cKO-livers was reduced to 44, 10, and 17% of WT-livers, respectively, indicating a broad decrease of HSPCs in cKO FLs (Fig. 1c). There were no differences noted between WT and Het (Fig. 1c, WT vs. Het).

De novo production of HSPCs in the embryo occurs within various hematopoietic sites, including the DA of the AGM, and extraembryonic sites, such as the YS, the placenta, and the VA + UA[1–3]. Since all the aforementioned hematopoietic sites[19] express Cdh5-Cre recombinase, the extent of HSPC defects due to Drosha depletion in each site was evaluated by CFU assay (Fig. 1d and Supplementary Fig. 2a). At E10.5, the number of colonies was significantly reduced in the AGM in cKO ((45% (CFU-GMs), 19% (BFU-Es), and 32% (CFU-GEMMs) of Ctr) (Fig. 1d). Furthermore, genomic DNA analysis ascertained that all the cKO AGM colonies (Fig. 1d) retained at least one intact Drosha allele (Supplementary Table 1), implying that Drosha-null HSPCs in the AGM are nonfunctional or undergo premature cell death. Reduction of HSPCs was also observed in the cKO extraembryonic hematopoietic sites as early as E9.5 (Supplementary Fig. 2a). Thus, Drosha depletion impairs the production of HSPCs from the HE.

Although at E12.5 there was no discernible developmental abnormalities, by E13.5 about half of the cKO embryos exhibited subcutaneous hemorrhages (Fig. 1a, *yellow arrow*). All cKO embryos died between E14.5 and E15.5 (Supplementary Table 2). Immunostaining analysis of CD31, an endothelial marker, of the YS vasculature of E13.5 cKO embryos revealed an ~20% reduction in blood vessel branching compared to Ctr (Supplementary Fig. 2b), suggesting that late endothelial patterning is mildly affected by the deletion of *Drosha*. At E14.5, 90% of cKO embryos presented both edema (Supplementary Fig. 1e, *red arrow*) and bilateral hemorrhages characteristic of lymphatic-vein

fusion defects[22] (Supplementary Fig. 1e, *blue arrow*). Histological analysis of cKO embryos verified blood accumulation in dilated jugular lymph sacs (Supplementary Fig. 2c) and the fusion of jugular lymph sac and internal jugular vein (Supplementary Fig. 2c, *red asterisk*). This phenotype is likely due to a lack of intact platelet function, which is required for proper separation of blood and lymphatic circulatory systems[23], and is commonly found in mouse models with hematopoietic defects, including *Gata2*[24], *Runx1*[25] and *Chd1*[26] mutant mice. On the other hand, heart rate (Supplementary Movie 1 for Ctr and 2 for cKO), circulation (Supplementary Movie 3 for Ctr and 4 for cKO), and overall size and morphology of the heart were indistinguishable between cKO and Ctr, with the exception of a thinner compact layer of the left ventricle in cKO embryos (Supplementary Fig. 2d). Thus, we conclude that the primary cause of lethality of cKO embryos is due to a hematopoietic defect.

**Normal endothelial to hematopoietic transition in Drosha mutants.** HSPCs emerge from the HE as hematopoietic clusters through EHT[7]. The appearance of clusters coincides with the acquisition of cell surface markers Kit (CD117)[27–30] and CD41[29, 31] in addition to the pan-EC marker CD31. Therefore, cell surface marker phenotypes (CD31$^+$Kit$^+$CD45$^-$ and CD31$^+$CD41$^+$CD45$^-$) are commonly used to define HSPC cluster populations[32]. To our surprise, flow cytometric analysis showed no change of HSPC cluster populations (CD31$^+$Kit$^+$CD45$^-$) and endothelial populations (CD31$^+$) in the cKO AGM at E10.5 (Fig. 2a, f, g) and E11.5 (Supplementary Fig. 3a, c). HSPC cluster populations defined by the markers CD31$^+$CD41$^+$CD45$^-$ also demonstrated no differences between cKO and Ctr at E10.5 (Supplementary Fig. 3b, e)[27, 28]. Unlike early HSPC population, a more mature HSPC population (CD31$^+$Kit$^+$CD45$^+$)[32] demonstrated a significant decrease in cKO at both E10.5 and E11.5 (Fig. 2a, h, 0.8% in cKO vs. 1.4% in Ctr and Supplementary Fig. 3a, d), suggesting that the maturation of HSPC clusters towards a hematopoietic (CD45$^+$) identity is compromised in cKO. Previous studies suggest that changes in the expression of Runx1, a transcription factor critical for EHT at E10.5 AGM[19, 32], could explain this phenotype. However, immunofluorescence (IF) staining revealed that HSPC clusters from cKO AGM express both

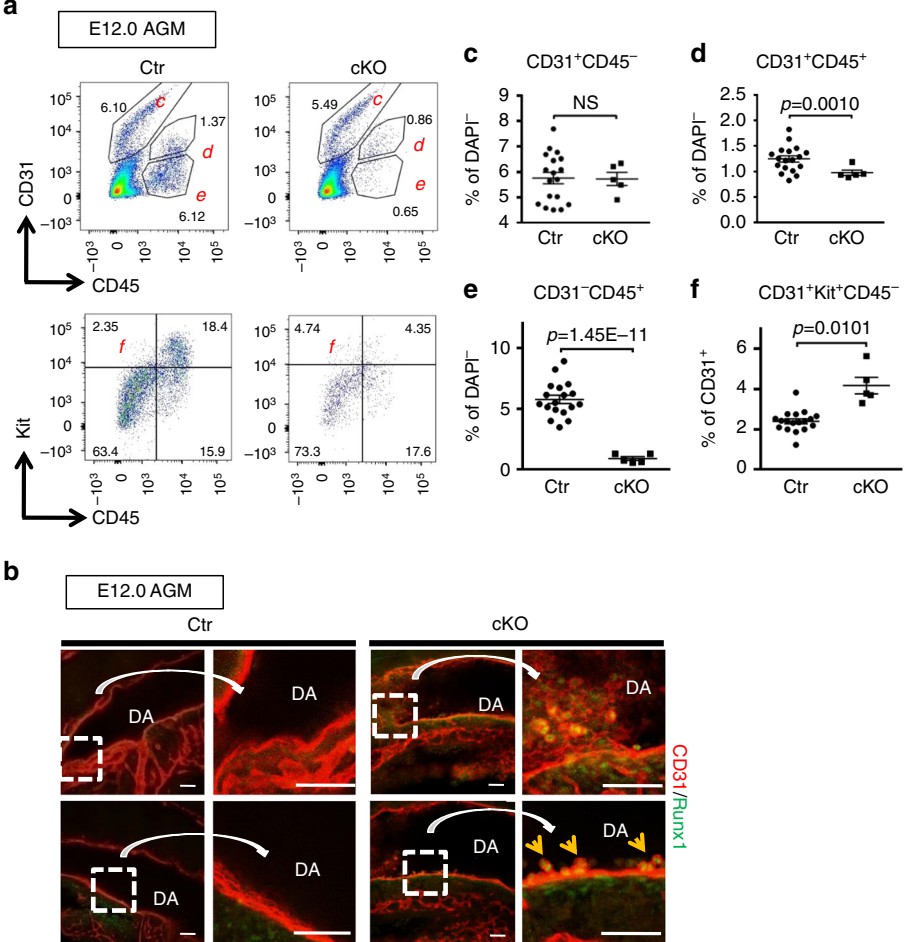

**Fig. 4** Accumulation of HSPC clusters in the AGM of Drosha mutants. **a** ECs (CD31+CD45−), HSPC clusters (CD31+Kit+CD45−), maturing HSPCs (CD31+CD45+), and mature hematopoietic population (CD31−CD45+) in E12.0 Ctr or cKO AGMs were analyzed by flow cytometry. For ECs (CD31+CD45−), maturing HSPCs (CD31+CD45+) and mature hematopoietic cells (CD31−CD45+); the fraction (%) of each population among live (DAPI−) cells was plotted as Mean ± SEM; p values were generated by unpaired Student's t test. For HSPC clusters, the fraction (%) of CD31+Kit+CD45−among CD31+ cells was plotted. n = Ctr: 18 embryos, cKO: 5 embryos. 3 litters. **b** Whole-mount IF images of HSPC cluster cells in AGMs from E12.0 Ctr or cKO embryos were shown. Runx1 (*green*) and CD31 (*red*). *Arrows* indicate HSPC clusters. Areas shown with a *dotted box* in left panels were magnified and presented in right. Scale bars: 50 μm. n = Ctr: 3 embryos, cKO: 2 embryos. 1 litter. NS, not significant

Runx1 (*green*) and CD31 (*red*) and are morphologically indistinguishable from Ctr (Fig. 2b). Furthermore, scanning electron microscopy images of the AGM (Fig. 2c, *top*) and quantitation of cluster cells (Fig. 2c, *bottom*) demonstrate that the size and the frequency of HSPC clusters are indistinguishable between cKO and Ctr. When the cKO mouse was crossed to the *Rosa26mTmG* Cre reporter mouse[20] to estimate Cre recombination rate, we found ~90% of ECs and cluster cells underwent successful recombination reported by GFP expression (Fig. 2d, *middle, arrow*). Furthermore, qRT-PCR analysis showed 94% reduction of *Drosha* mRNA in HSPC clusters (CD31+Kit+CD45−) (Fig. 2d, *right*). Therefore, normal morphology and quantity of cluster cells found in cKO AGMs is not due to incomplete excision of *Drosha* loci.

When ECs sorted from E10.5 AGM are co-cultured with mouse stromal OP9-DL1 cells, a fraction of ECs undergo EHT to generate HSPC clusters in vitro[33, 34]. Using this OP9 co-culture system[33, 34], we compared the ability of ECs (CD31+) from cKO or Ctr to produce HSPC clusters. As observed in vivo (Fig. 2c), the morphology (Fig. 2e, *top*) and the number of clusters (Fig. 2e, *bottom left*) derived from cKO-ECs were indistinguishable from the clusters from Ctr-ECs. However, when the clusters were subjected to CFU assays, cKO ECs-derived clusters gave rise to

fewer colonies as compared to the clusters from Ctr-ECs (14 vs. 79, Fig. 2e, *bottom right*). These findings indicate that HSPC clusters emerge normally from cKO AGMs, however, they are functionally defective. Taken together, these data support that Drosha activity is dispensable for the specification of HE and EHT, but is required at a later stage of HSPC maturation.

**Defects of HSPC maturation in Drosha mutants**. To examine the maturation defect of HSPC clusters in cKO at later developmental stage, cells from E11.5 AGM were subjected to flow cytometric analysis. CD45+ hematopoietic populations were reduced in cKO AGMs at E11.5 (CD31+CD45+ (Fig. 3a, d, 0.94% in cKO vs. 2.11% in Ctr; and Supplementary Fig. 4a, d) or Kit+CD45+ (Fig. 3a, f, 2.90% in cKO vs. 6.64% in Ctr)), and non-cluster hematopoietic populations (CD31−CD45+) were also depleted in cKO at E11.5 (Fig. 3a, e, 0.4% in cKO vs. 2.0% in Ctr, Supplementary Fig. 4a, e). To evaluate the function of the hematopoietic populations, the same number of CD45+ cells was isolated from E11.5 AGM or FL of Ctr or cKO and subjected to CFU assays. The number of colonies from cKO AGM and FL were ~7-fold and 10-fold smaller than Ctr, respectively (Fig. 3b).

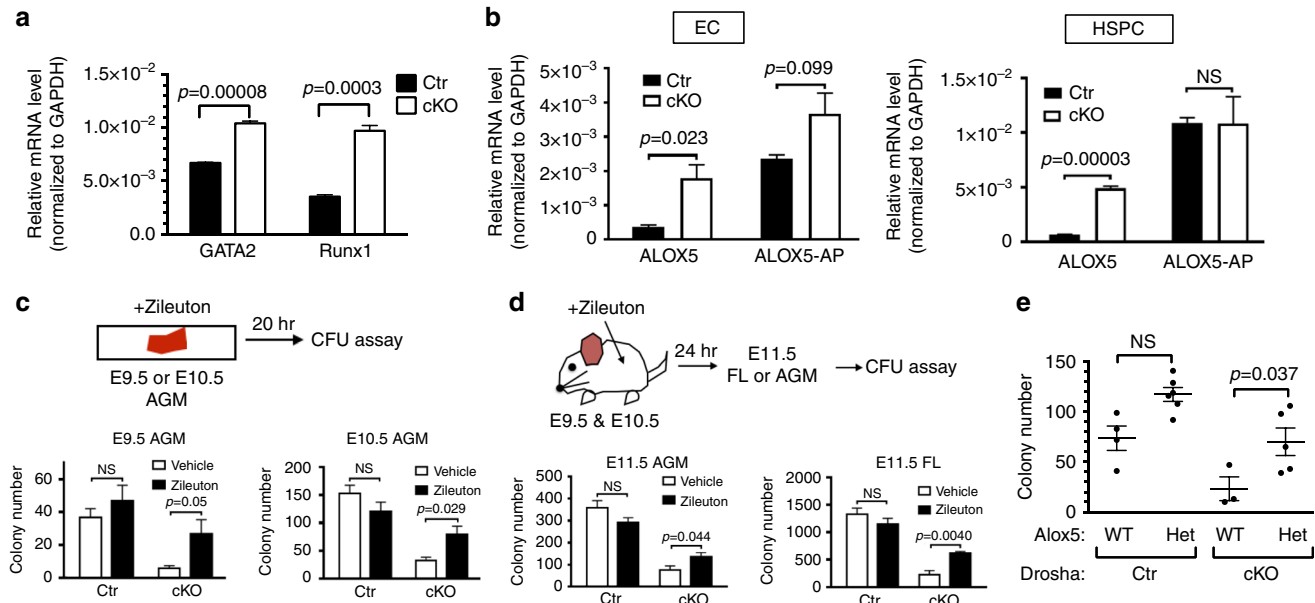

**Fig. 5** Inhibition of Alox5 rescues the phenotype in Drosha cKO embryos. **a** A modest increase of *Gata2* and *Runx1* mRNAs relative to *GAPDH* in HSPCs (CD31[+]Kit[+]CD45[−]) from E11.5 cKO embryos compared to Ctr embryos were found by qRT-PCR analysis in triplicates (Mean ± SEM); *p* values were generated by unpaired Student *t*-test. Ctr: 12 embryos, cKO: 6 embryos. *n* = 3 litters. **b** qRT-PCR analysis of *Alox5* and *Alox5-AP* mRNAs relative to *GAPDH* in ECs (CD31[+]Kit[−]CD45[−], *left*) and HSPC clusters (CD31[+]Kit[+]CD45[−], *right*) from E11.5 Ctr or cKO embryos is shown as Mean ± SEM; *p* values were generated by unpaired Student *t*-test. *n* = Ctr: 12 embryos, cKO: 6 embryos. 3 litters. **c** Schematic diagram of ex vivo zileuton rescue experiment (top panel). AGMs isolated from E9.5 (*bottom left*) or E10.5 (*bottom right*) Ctr or cKO embryos were incubated with 50 μM zileuton or vehicle for overnight, and all cells from AGMs were subjected to CFU assay. Total counts of colonies (BFU-E, CFU-GM, and CFU-GEMM) were presented as Mean ± SEM; *p* values were generated by unpaired Student's *t* test . For zileuton treatment, *n* = Ctr: 11 embryos, cKO: 6 embryos. 3 litters for E9.5 and *n* = Ctr: 18 embryos, cKO: 9 embryos. 5 litters for E10.5. For vehicle, *n* = Ctr: 11 embryos, cKO: 3 embryos. 3 litters for E9.5 and *n* = Ctr: 18 embryos, cKO: 10 embryos. 6 litters for E10.5. **d** Schematic diagram of in vivo zileuton rescue experiment (top panel). Zileuton or vehicle 5 mg kg[−1] was injected into pregnant mice at E9.5 and E10.5. At E11.5, AGM and FL were harvested and all cells from AGMs (*bottom left*) and FL (*bottom right*) were subjected to CFU assay in duplicates. Total counts of colonies (BFU-E, CFU-GM, and CFU-GEMM) were presented as Mean ± SEM; *p* values were generated by unpaired Student's *t* test . For zilueton, *n* = Ctr: 11 embryos, cKO: 6 embryos. 4 litters. For vehicle, *n* = Ctr: 10 embryos, cKO: 5 embryos. 3 litters. **e** One percent of cells from E13.5 FLs were subjected to CFU assay. Total counts of colonies (BFU-E, CFU-GM, and CFU-GEMM) were presented as Mean ± SEM. Results were plotted as Mean ± SEM; *p* values were generated by unpaired Student's *t* test . Drosha Ctr stands for Drosha Ctr (*Drosha*[fl/fl] or *Drosha*[fl/+] or *Drosha*[fl/+];*Cdh5-Cre*[+]) and Drosha cKO stands for Drosha cKO (*Drosha*[fl/fl];*Cdh5-cre*[+]). Alox5 WT and Het stand for *Alox5*[+/+] and *Alox5*[+/−], respectively. *Drosha* Ctr;*Alox5*[+/+]: *n* = 9 embryos. *Drosha* Ctr;*Alox5*[+/−]: *n* = 8 embryos. *Drosha* cKO;*Alox5*[+/+]: *n* = 3 embryos. *Drosha* cKO;*Alox5*[+/−]: *n* = 4 embryos. 5 litters. NS, not significant

Furthermore, genomic DNA analysis confirmed that all the colonies derived from cKO AGM or FL retained at least one intact *Drosha* allele (Supplementary Table 3), indicating that Drosha-null CD45[+] cells are unable to produce hematopoietic colonies. Therefore, Drosha activity is essential for the functional maturation of HSPCs.

**Immobilization of HSPC clusters in the mutant AGM.** A smaller number of maturing HSPCs and hematopoietic cells in cKO could arise from abnormal survival or proliferation of HSPC clusters[26, 35]. However, no evidence of premature cell death (Supplementary Fig. 5a–c) or a proliferation defect (Supplementary Fig. 5d, e) of HSPC clusters in cKO was found. Thus, we shifted the attention to HSPC clusters towards the end of the HE time window, E12. To our surprise, there existed approximately a 2-fold increase in HSPC cluster populations (CD31[+]Kit[+]CD45[−]) in the cKO AGMs compared to Ctr (Fig. 4a, f) at E12. However the maturing HSPC (CD31[+]CD45[+]) (Fig. 4a, d) and hematopoietic populations (CD31[−]CD45[+]) (Fig. 4a, e) were significantly reduced in the cKO AGMs to 65 and 12% of Ctr, respectively. This result suggests an attenuation of the maturation toward CD45[+] cells in cKO HSPC clusters. Furthermore, IF analysis of cluster cells detected large HSPC clusters in the caudal region of E12 cKO AGMs (Fig. 4b, *yellow arrows*). No clusters were detected in Ctr at E12 (Fig. 4b). These data indicate that HSPC

clusters are immobilized in the AGM of cKO, with a failure of maturation and inability to migrate to the FL.

**Aberrant activation of the LT biosynthesis pathway in Drosha mutants.** To identify the molecule(s) involved in the maturation defect of cKO HSPCs, we performed a comparative transcriptome analysis using ECs (CD31[+]Kit[−]CD45[−]) and HSPC clusters (CD31[+]Kit[+]CD45[−]) sorted from 8 WT (*Drosha*[fl/+] or *Drosha*[fl/fl]), 5 Het (*Cdh5-Cre*[+]; *Drosha*[fl/+]) and 5 cKO (*Cdh5*[−]*Cre*[+]; *Drosha*[fl/fl]) at E11.5 (Supplementary Fig. 4b). As global depletion of miRNAs leads to a de-repression of their targets, we noted that the majority of genes expressed in ECs and HSPCs were increased in cKO compared to Ctr. The amounts of various EC markers in ECs and HSPCs from cKO were similar or slightly higher than Ctr (Supplementary Data 1), further demonstrating that EC identity is intact in cKO. Similarly, the amounts of critical factors regulating EHT, such as Runx1 and GATA2, were mildly increased in cKO (Supplementary Data 1). qRT-PCR analysis confirmed the RNAseq data, indicating that *Gata2* and *Runx1* mRNAs were increased 1.5-fold and 2.7-fold in cKO HSPCs compared to Ctr HSPCs, respectively (Fig. 5a). Reduction of Gata2 gene dosage[36] in the cKO mice (*Drosha*[fl/fl];*Gata2*[fl/+];*Cdh5-Cre*[+]) did not ameliorate hematopoietic defects (Supplementary Fig. 6), suggesting that the elevation of *GATA2* is not a primary cause of the cKO phenotype.

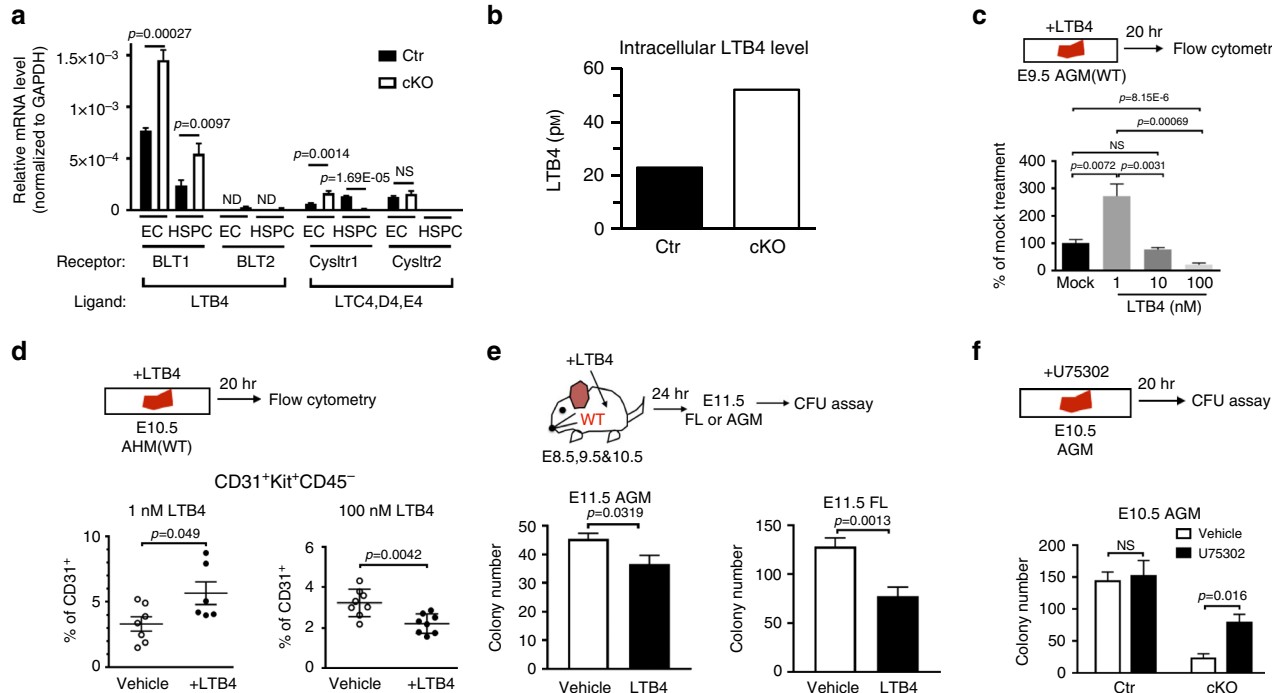

**Fig. 6** Increased LTB4 synthesis leads to HSPC defects in Drosha cKO embryos. **a** mRNA levels of LT receptors in ECs (CD31$^+$Kit$^-$CD45$^-$) and HSPCs (CD31$^+$Kit$^+$CD45$^-$) from Ctr or cKO embryos. Mean ± SEM. $n =$ Ctr: 12 embryos, cKO: 6 embryos. 3 litters. **b** HSPCs (CD31$^+$Kit$^+$CD45$^-$) from E11.5 AGM were subjected to ELISA analysis, and the concentration of LTB4 (pM) was plotted as Mean ± SEM. $n =$ Ctr: 6 embryos, cKO: 4 embryos. 3 litters. **c** E9.5 WT AGMs were incubated with LTB4 or vehicle for 20 h ex vivo, followed by CFU assay (bottom panel). Relative colony number was plotted as Mean ± SEM; $p$ values were generated by unpaired Student's $t$ test. $n = 8$ embryos (vehicle) and 9 embryos (LTB4), 3 litters for 1 nM LTB4. $n = 7$ embryos (vehicle) and 8 embryos (LTB4), 2 litters for 10 nM LTB4. $n = 8$ embryos (vehicle) and 7 embryos (LTB4), 2 litters for 100 nM LTB4. **d** Flow cytometric analysis on E10.5 WT AGMs treated with 1 nM LTB4 (bottom left) or 100 nM LTB4 (bottom right). The fraction (%) of CD31$^+$Kit$^+$CD45$^-$ cells among CD31$^+$ cells was plotted as Mean ± SEM; $p$ values were generated by unpaired Student's $t$ test. $n = 7$ embryos (vehicle) and 6 embryos (LTB4), 2 litters for 1 nM LTB4. $n = 8$ embryos (vehicle) and 7 embryos (LTB4), 2 litters for 100 nM LTB4. **e** LTB4 or vehicle was injected to pregnant WT mice at E8.5, E9.5, and E10.5. 20 and 10% of cells from AGM and FL from E11.5 embryo were subjected to CFU assay, respectively. Colony number was plotted as Mean ± SEM; $p$ values were generated by unpaired Student's $t$ test. $n = 10$ embryos from 2 vehicle-injected litters, $n = 10$ embryos from 2 LTB4-injected litters. **f** E10.5 Ctr or cKO AGMs were incubated with U75302 or vehicle, followed by CFU assay. Colony number was plotted as Mean ± SEM; $p$ values were generated by unpaired Student's $t$ test. For U75302 treatment, $n =$ Ctr: 5 embryos, cKO: 4 embryos. 3 litters. For vehicle, $n =$ Ctr: 7 embryos, cKO: 4 embryos. 2 litters

To identify other factors possibly responsible for the cKO phenotype, we screened the RNAseq data for transcripts detectable in both cKO and Ctr ((>5 reads per kilobase per million reads (RPKM)) but differentially expressed in the two samples ($p < 0.001$). This group included three genes involved in the arachidonic acid metabolic pathway: *phospholipase A2 group VI* (*Pla2g6*), *Alox5*, and its cofactor *Alox5-activating protein* (*Alox5-AP*) (Supplementary Data 1). *Pla2g6* mRNA, which encodes an enzyme that releases arachidonic acid from phospholipids[37], was increased ~3-fold in cKO (Supplementary Data 1). The Alox5/Alox5-AP is a key enzyme complex in leukotriene (LT) biosynthesis that catalyzes the conversion of arachidonic acid to the unstable intermediate product LTA4, which is further metabolized to more stable LTs (LTB4, LTC4, LTD4, and LTE4)[38, 39]. qRT-PCR confirmed higher amounts of *Alox5* mRNA in cKO ECs (CD31$^+$CD45$^-$) and HSPCs (CD31$^+$Kit$^+$CD45$^-$) (Fig. 5b). Thus, we hypothesized that augmented production of LTs, the products of Alox5, in cKO HSPCs might be responsible for the observed hematopoietic defects. To test this hypothesis, AGM explants from E9.5, E10.5 cKO or Ctr were treated with a pharmacological inhibitor of Alox5, zileuton (A-64077), followed by CFU assays[40, 41]. Zileuton is a clinically approved drug to treat asthma that inhibits LT production by eosinophils[42, 43]. Vehicle-treated E9.5 AGMs from cKO produced on average 6 colonies, while zileuton treatment increased colony formation 4.5-fold, to 27 colonies (Fig. 5c and

Supplementary Fig. 7a). Similarly, zileuton treatment of E10.5 AGMs increased the number of HSPCs 2.4-fold, from 33 colonies to 80 colonies (Fig. 5c and Supplementary Fig. 7b). Thus, pharmacological inhibition of Alox5 at E9.5 or E10.5 rescues hematopoietic defects in cKO AGMs. Next we tested the rescue effect of zileuton in vivo. Zileuton or vehicle was injected into pregnant mice at E9.5 and E10.5. At E11.5, AGMs and FLs were isolated from the embryos and subjected to CFU assays. AGMs and FLs from cKO treated in vivo with zileuton contained 1.7-fold and 2.8-fold more colonies, respectively, compared to vehicle-treated cKO tissues (Fig. 5d and Supplementary Fig. 7c, d). Furthermore, the size of the liver became significantly increased by zileuton treatment in cKO (Supplementary Fig. 7e). Therefore, inhibition of the Alox5-LT pathway partially rescues the hematopoietic defects in cKO. Zileuton treatment of Ctr animals resulted in mild reduction of HSPC numbers in the AGM and the FL compared to vehicle treatment (Fig. 5d), and a similar trend was observed when E10.5 AGMs were treated with zileuton ex vivo (Fig. 5c, bottom right). These results suggest that tight controls of LT levels are prerequisite for normal hematopoiesis in wild-type mice. To exclude the possibility of off-target effects by zileuton, cKO mice were crossed with *Alox5* heterozygous null (Het) mice to generate *Drosha*:cKO/*Alox5*:Het mice, in which the amount of Alox5 mRNA was reduced by ~50% (Supplementary Fig. 7f, left). Consistent with the results of zileuton treatment (Fig. 5d), cKO with reduced *Alox5* gene dosage exhibited 2.3-fold

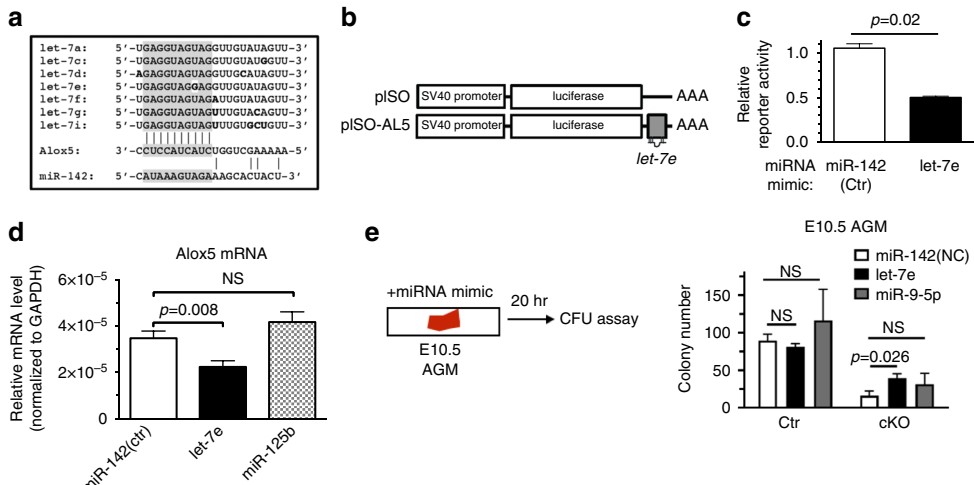

**Fig. 7** Alox5 is a novel target of the *let-7* family of miRNAs. **a** Sequence comparison of the *let-7* family members and the sequence partially complementary to the seed sequence of *let-7* (*shaded*) found in the 3′-UTR of *Alox5* mRNAs. **b** Schematic diagram of the *Alox5*-reporter construct (*pISO-AL5*) in which let-7 binding sequence in the *Alox5* 3′UTR (*hatched box*) was cloned at the 3′ end of the firefly luciferase gene upstream of SV40 poly(A). **c** pISO-AL5 or empty pISO vector was cotransfected into HEK293 cells with 10 nM *let-7e* or *miR-142* (control) mimic or vehicle alone (*bottom*). Relative activities of the reporter were shown as a ratio of pISO-AL5/pISO (Mean ± SEM); p values were generated by unpaired Student's t test . n = 6. **d** Macrophage RAW 264.7 cells were transfected with 18 nM *miR-142* (control), *let-7e*, or *miR-125b* mimic, followed by qRT-PCR analysis of *Alox5* mRNAs (*bottom*). Results were plotted as Mean ± SEM; p values were generated by unpaired Student's t test. **e** AGMs from E10.5 Ctr or cKO embryos were transfected with miRNA, followed by CFU assay. Colony number was plotted as Mean ± SD (*right*); p values were generated by unpaired Student's t test . For *miR-142*, n = Ctr: 9 embryos, cKO: 5 embryos. 2 litters. For *let-7e*, n = Ctr: 23 embryos, cKO: 6 embryos. 4 litters. For *miR-9-5p*, n = Ctr: 7 embryos, cKO: 4 embryos. 2 litters. NS, not significant. ND, not detectable

more HSPCs than cKO with two *Alox5* loci (*Drosha*:cKO/*Alox5*: WT) (Fig. 5e, and Supplementary Fig. 7g). Thus, aberrant activation of the Alox5-LT pathway in cKO embryos appears to hamper the maturation of HSPCs, which can be rescued by either pharmacological or genetic intervention of *Alox5* dosage.

**Increased LTB4 impairs the release of HSPC clusters**. As a step toward identifying which LT is responsible for the hematopoietic defect in cKO, we examined the transcripts of BLT1, a high-affinity receptor for LTB4; BLT2, a low-affinity receptor for LTB4; and Cysltr1 and Cysltr2, receptors for cysteinyl LTs (LTC4, LTD4, and LTE4)[44] in HSPCs. *BLT1* mRNA was abundant in HSPCs and ECs, and ~2-fold higher in cKO as compared to Ctr (Fig. 6a). *BLT2, Cysltr1* and *Cysltr2* mRNAs were less abundant compared to *BLT1* (Fig. 6a). Next, we quantitated the intracellular LTB4 in HSPCs from E10.5 AGMs by an enzyme-linked immunosorbent assay (ELISA), to find a concentration of LTB4 of 52 pM in cKO and 23 pM in Ctr (Fig. 6b). This result demonstrates that LTB4 production is augmented in cKO HSPCs, which is in agreement with the increased amount of Alox5 detected in cKO (Fig. 5b). To test the effect of LTB4 on HSPCs, E9.5 AGMs from wild-type (WT) embryos were treated with increasing concentrations of LTB4, and subjected to CFU assays (Fig. 6c, *upper panel*). The treatment of WT AGMs with 1 nM LTB4 augmented the number of colonies by 2.7-fold compared to vehicle-treatment (Fig. 6c). On the contrary, 100 nM LTB4 reduced the number of colonies to 25% of vehicle-treated control (Fig. 6c). Consistent with the result of CFU assays, flow cytometric analysis of HSPC (CD31+Kit+CD45−) populations revealed a 70% increase upon 1 nM LTB4 (Fig. 6d, *left*), and 68% decrease upon 100 nM LTB4 treatment (Fig. 6d, *right panel*). Thus, the effect of LTB4 on AGMs is dose-dependent. Next, the in vivo effects of LTB4 were tested by administering LTB4 or vehicle by intraperitoneal injection to WT pregnant female mice between E8.5 and E10.5. E11.5 AGMs and FLs were isolated from the embryos and tested in CFU assays. Compared to vehicle-

treated WT, LTB4-treated WT produced 20% less colonies from the AGM (Fig. 6e), and even less (40% reduction) from the FL (Fig. 6e). These results indicate that exposure to increased amounts of LTB4 causes impairment of colony forming ability and a reduction of HSPCs, which together indicate a significant role for LTB4 biosynthesis and homeostasis during developmental hematopoiesis. To ameliorate the effect of aberrant intracellular LTB4 signaling in cKO HSPCs, E10.5 AGMs were treated with a small molecule inhibitor of the BLT1 receptor, U75302[45], and subjected to CFU assays. U75302 treatment increased the colony number 3.4-fold (from 23 to 79) in cKO (Fig. 6f and Supplementary Fig. 7h). Therefore, Drosha activity is essential to restrain the intracellular LTB4 signaling pathway in HSPCs during development.

**Depletion of Let-7 increases Alox5**. To identify the miRNAs that might be responsible for the induction of *Alox5* mRNA, we isolated miRNAs from ECs (CD31+ CD45−Kit−) of E11.0 Ctr or cKO AGMs (Supplementary Fig. 4b). As expected, ~82% of miRNAs (261 out of 320 miRNAs) detected in Ctr-ECs were decreased in cKO AGMs by >1.6-fold in terms of reads per million (RPM), supporting the essential role of Drosha in miRNA biosynthesis (Supplementary Data 2). Thirty-one percent of miRNAs present in Ctr AGMs (81 out of 261) were undetectable in cKO AGMs, and an additional 21% (54 out of 261) were reduced more than 50% in cKO AGMs (Supplementary Data 2). The *let-7* family of miRNAs, which shares a common seed sequence (Fig. 7a), was abundant in Ctr AGMs, and reduced more than 50% in cKO AGMs (Supplementary Table 4). The 3′-untranslated region (UTR) of the *Alox5* mRNA contains a predicted *let-7* binding sequence (Fig. 7a). When *Alox5* 3′-UTR was inserted into the luciferase reporter construct pISO-AL5 (Fig. 7b), the luciferase activity was significantly reduced upon transfection of *let-7e* mimic but not control mimic (*miR-142*) (Fig. 7c). Transfection of *let-7e* mimic but not *miR-125* or *miR-142* mimic into RAW 264.7 cells mediated the reduction of *Alox5* mRNA

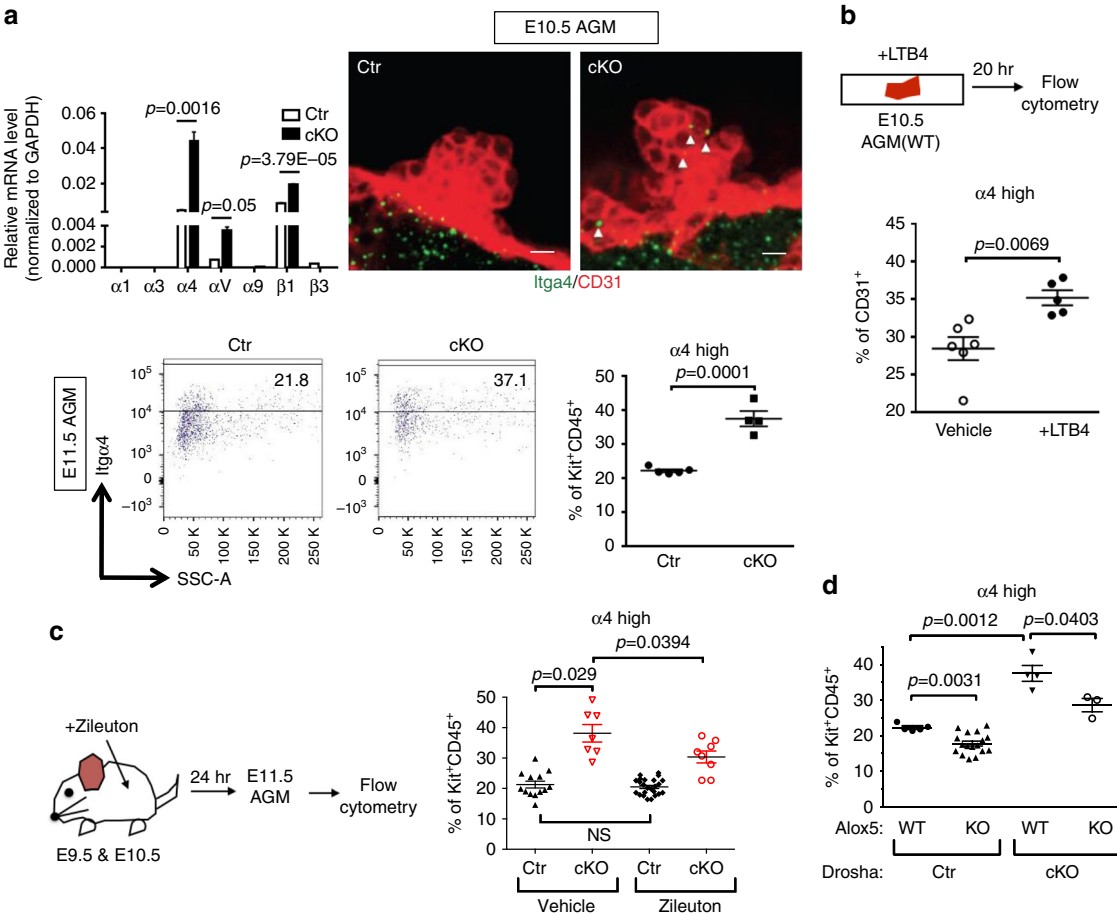

**Fig. 8** Elevation of Integrin α4β1 in HSPCs from Drosha cKO embryos. **a** qRT-PCR analysis of *Itgα1, Itgα3, Itgα4, ItgαV, Itgα9, Itgβ1*, and *Itgβ3* mRNAs relative to *GAPDH* in HSPC clusters (CD31⁺Kit⁺CD45⁻) from E11.5 Ctr or cKO embryos (*top left*). Results were plotted as Mean ± SEM; *p* values were generated by unpaired Student's *t* test. *n* = Ctr: 12 embryos, cKO: 6 embryos. 3 litters. Whole-mount IF staining of Itgα4 (*green*) and CD31 (*red*) of E10.5 AGM from cKO or Ctr embryos was shown (*middle*). *Arrow heads*: α4-positive cells in HSPC clusters (*top right*). Lower panels: flow cytometric analysis of the fraction (%) of Itgα4 high populations among Kit⁺CD45⁺ cells in E11.5 AGMs from Ctr or cKO embryos. *n* = Ctr: 5 embryos, cKO: 4 embryos. 2 litters. Scale bars: 10 μm. **b** Schematic diagram of ex vivo LTB4 treatment experiment (*top*). At E10.5, AGMs from WT embryos were harvested and incubated with 300 nM LTB4 or vehicle for 20 h, followed by flow cytometry. The fraction (%) of Itgα4 high cells in endothelial population (CD31⁺) was plotted as Mean ± SEM (*bottom*); *p* values were generated by unpaired Student's *t* test . *n* = 7 embryos for vehicle and *n* = 7 embryos for LTB4. 2 litters. **c** Schematic diagram of in vivo zileuton treatment experiment (left panel). Results were plotted as Mean ± SEM; *p* values were generated by unpaired Student's *t* test. Zileuton or vehicle (5 mg kg⁻¹) was injected into pregnant mice at E9.5 and E10.5. At E11.5, AGMs were harvested and subjected to flow cytometry to quantitate the fraction (%) of Itgα4 high populations among Kit⁺CD45⁺ cells. For zileuton, *n* = Ctr: 24 embryos, cKO: 8 embryos. 3 litters. For vehicle, Ctr: 13 embryos, cKO: 7 embryos. *n* = 4 litters. **d** Flow cytometric analysis was performed to quantitate the fraction (%) of Itgα4 high HSPCs among Kit⁺CD45⁺ cells in E11.5 AGMs from *Drosha* Ctr;*Alox5* WT, *Drosha* Ctr;*Alox5* KO, *Drosha* cKO;*Alox5* WT or *Drosha* cKO;*Alox5* KO embryos. Results were plotted as Mean ± SEM; *p* values were generated by unpaired Student's *t* test. *Drosha* Ctr;*Alox5* WT *n* = 5 embryos. *Drosha* Ctr;*Alox5* KO *n* = 17 embryos. *Drosha* cKO;*Alox5* WT *n* = 4 embryos. *Drosha* cKO;*Alox5* KO *n* = 3 embryos. 6 litters

(Fig. 7d), indicating that *Alox5* is a novel target of the *let-7* family of miRNAs. Therefore, we hypothesized that restoration of *let-7* might be sufficient to rescue hematopoietic defects in cKO. *Let-7e* mimic, *miR-9-5p* mimic, or control mimic (*miR-142*) was transfected into E10.5 AGMs from cKO or Ctr, followed by CFU assays (Fig. 7e, *left*). *miR-9-5p* is known to target *Runx1*[46] and was detectable in HSPCs (Supplementary Table 4). The transfection of *miR-9-5p* mimic into cKO AGMs did not change colony counts compared to control mimic-transfected cKO AGMs (Fig. 7e, *right*), suggesting that the restoration of *miR-9-5p* and down-regulation of *Runx1* is not sufficient to rescue HSPC defect in cKO embryos. When *let-7e* was transfected, however, the number of total colonies increased 2.5-fold compared to control, demonstrating that the restoration of *let-7e* and downregulation of *Alox5* is sufficient to rescue the functional maturation of Drosha-depleted HSPCs.

**Increased α4β1 integrin expression in Drosha mutant HSPCs.** Comparative transcriptome analyses showed that several members of the integrin (Itg) family were upregulated in cKO HSPCs compared to Ctr HSPCs (Supplementary Data 1). qRT-PCR analyses confirmed that *Itgα4, ItgαV,* and *Itgβ1* mRNAs were elevated in cKO HSPCs. In particular, *Itgα4* mRNA was 8.4-fold more abundant in cKO than Ctr (Fig. 8a, *top left*). Immunofluorescence also confirmed higher amounts of Itgα4 protein in the HE clusters in cKO (Fig. 8a, *top right*). While flow cytometric analysis of cell surface expression of Itgα4 in HSPCs (CD31⁺Kit⁺CD45⁻) did not uncover significant differences between cKO and Ctr (Supplementary Fig. 8a), maturing HSPC populations (Kit⁺CD45⁺) demonstrated a 1.7-fold higher fraction of Itgα4 high cells (α4 high cells) in cKO as compared to Ctr (Fig. 8a, *bottom*). When WT AGMs were treated with 300 nM LTB4 ex vivo, a 1.3-fold larger fraction of α4 high cells was

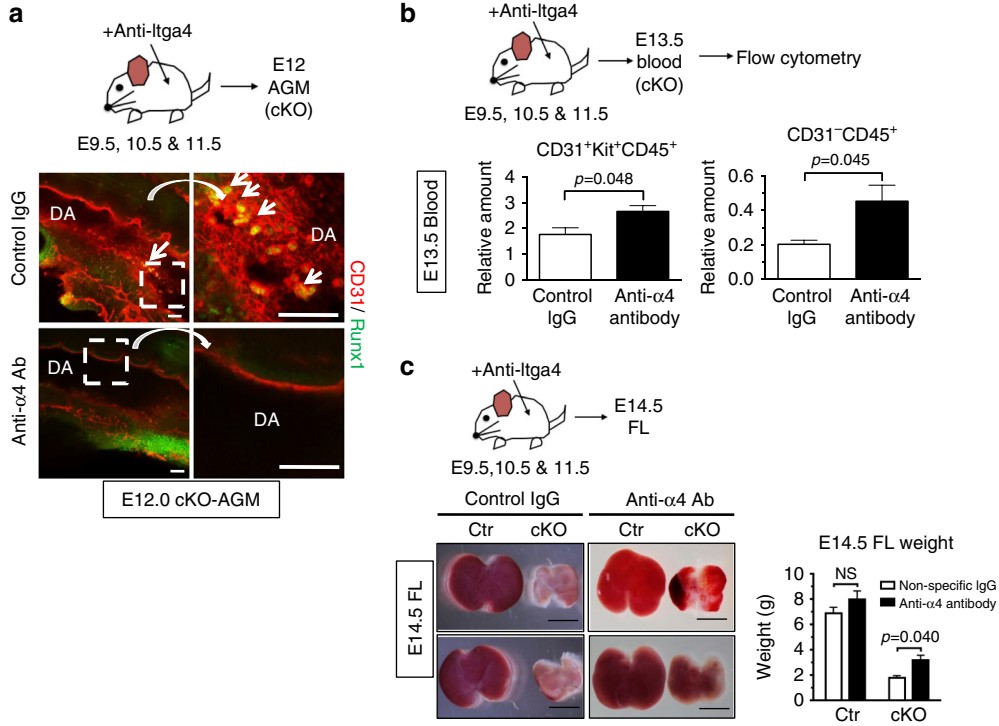

**Fig. 9** Inhibition of Integrin α4 in Drosha cKO embryo permits mobilization of HSPCs and transition to liver. **a** Schematic diagram of in vivo anti-Itgα4 antibody injection experiment (top panel). Anti-Itgα4 neutralizing antibody, 2 mg kg$^{-1}$ or non-specific IgG (control) was injected into pregnant mice at E9.5, E10.5, and E11.5. At E12.0, cKO-AGMs from anti-Itgα4 antibody- or control IgG-injected litters were subjected to whole-mount IF analysis of Runx1 (*green*) and CD31 (*red*) in HSPC cluster (*arrows*). Areas of the *dotted box* in left panels were magnified and shown in right. DA: DA. *Scale bars*: 50 μm. n = Ctr: 3 embryos, cKO: 2 embryos. 1 litter. **b** Schematic diagram of in vivo anti-Itgα4 antibody rescue experiment (*top*). Anti-Itgα4 antibody, 2 mg kg$^{-1}$ or non-specific IgG (control) was injected into pregnant mice at E9.5, E10.5 and E11.5, then the peripheral blood was collected from E13.5 embryos and subjected to flow cytometry (*bottom right*). Relative amount of HSPC (CD31$^+$Kit$^+$CD45$^+$) (*bottom left*) or mature hematopoietic cell (CD31$^-$CD45$^+$) (*bottom right*) populations in the peripheral blood from E13.5 cKO embryos was plotted as Mean ± SEM; p values were generated by unpaired Student's t test . For control IgG, n = cKO: 3 embryos. 2 litters. For anti-Itgα4 antibody, n = cKO: 6 embryos. 3 litters. **c** *Schematic diagram* of in vivo anti-Itgα4 antibody rescue experiment (*top*). anti-Itgα4, 2 mg kg$^{-1}$ or non-specific IgG (control) was injected into pregnant mice at E9.5, E10.5, and E11.5. At E14.5, FLs were harvested from Ctr or cKO embryos, weighed and plotted as Mean ± SEM (right panel); p values were generated by unpaired Student's t test. Representative E14.5 FL images were presented (*bottom left*). For control IgG, n = Ctr: 5 embryos, cKO: 5 embryos. n = 3 litters. For anti-Itgα4 antibody, n = Ctr: 5 embryos, cKO: 4 embryos. 2 litters. NS, not significant. *Scale bars*: 2 mm

detected compared to vehicle-treated AGMs, suggesting that α4$^{high}$ populations can be induced by exposure to high levels of LTB4 (Fig. 8b). When E11.5 cKO AGMs were treated with zileuton to inhibit LTB4 production, the fraction of α4$^{high}$ cells was notably reduced from 38% (vehicle) to 30% (zileuton) (Fig. 8c). Similarly, genetic ablation of *Alox5* in cKO also reduced α4$^{high}$ cell populations from 37% (*Drosha*:cKO/*Alox5*:WT) to 30% (*Drosha*:cKO/*Alox5*:KO) (Fig. 8d). It is also notable that homozygous deletion of *Alox5* in mice with wild-type Drosha alleles (*Drosha*:Ctr/*Alox5*:KO) led to a mild reduction of α4$^{high}$ cell populations from 22 to 18% (Fig. 8d). Therefore, activation of the Alox5-LTB4 pathway in HSPCs results in an increase of α4$^{high}$ cell populations.

**Mobilization of HSPC clusters by inactivating α4 integrin.** Integrins function as receptors for various cell surface proteins and are key players in cell adhesion to other cells and to the extracellular matrix[47]. Therefore, we hypothesized that the overabundance of Itgα4 on the surface of cKO HSPCs might act to immobilize HSPC clusters by increased adhesion to the vascular wall, preventing release into the circulation and subsequent colonization of the FL. To test the hypothesis, a neutralizing antibody against Itgα4 (anti-Itgα4)[48] or non-specific (control) IgG was injected into pregnant mice at E9.5-E11.5, and HSPC

clusters in cKO AGMs were examined at E12.0 (Fig. 9a). As shown in Fig. 4b, aggregated HSPC clusters were detected in the DA of E12.0 cKO AGMs treated with control IgG (Fig. 9a, *arrows*), while at this time no clusters could be detected in Ctr. However, these cluster aggregates were absent in cKO AGMs treated with anti-Itgα4 antibody (Fig. 9a), suggesting a successful release of cluster cells from the AGM. Genetic depletion of Alox5 in cKO cleared the large clusters from the AGM (Supplementary Fig. 8b). Concurrently, 1.8-fold and 2.2-fold increases in maturing HSPCs (CD31$^+$Kit$^+$CD45$^+$) and hematopoietic cells (CD31$^-$CD45$^+$) were detected in the peripheral blood from anti-Itgα4 antibody-treated cKO compared to control IgG-treated cKO at E13.5, respectively (Fig. 9b). Thus, inactivation of Itgα4 partially restores the release of cluster cells into the circulation. Furthermore, FLs from E14.5 cKO exhibited more intense red coloration, accompanied by a 1.7-fold increase of FL weight, after anti-Itgα4 antibody treatment (Fig. 9c), indicative of improved migration of HSPCs to the FL. Consistently, anti-Itgα4 antibody treatment in cKO embryos resulted in increased numbers of CFUs in the AGM, FL, YS vasculature, UA + VA, and placenta (Supplementary Fig. 8c), suggesting that the inhibition of Itgα4 restores lineage commitment of HSPCs concurrent with their mobilization from the vascular niche. Although not statistically significant, a small increase of liver weight in Ctr was observed upon anti-Itgα4 antibody treatment (Fig. 9c, *right*). This may suggest that

inhibition of Itgα4 facilitates mobilization of HSPCs to the FL in controls, as well. Altogether, our study sheds light on a previously unappreciated role of miRNA-dependent control of innate LTB4 signal as a gatekeeper of the transition from the vascular niche to the FL niche.

## Discussion

MiRNAs have been implicated in different aspects of hematopoiesis, including maintenance of adult bone marrow HSC self-renewal and terminal differentiation of adult progenitors into various lineages[49]. Outside the context of in vitro culture systems or adult hematopoiesis[49], however, little is known about the roles of miRNAs in the hematopoietic system, especially during its early development. Our study of conditional *Drosha* knockout mice uncovers a role for *let-7* miRNAs in developmental hematopoiesis, within a newly discovered stage that occurs after the emergence of HSPC clusters but prior to their mobilization to the FL. Furthermore, our study sheds light on the physiological requirement of a fine-tuned LTB4 signaling pathway for the maturation of HSPCs, and control of their release from the AGM microenvironment.

Critical roles of miRNAs in adult hematopoiesis, such as contributing to the proper formation of the HSC niche in bone marrow, have been gleaned from an osteoprogenitor-specific Dicer deletion, which presents with peripheral blood cytopenia and myelodysplasia[50]. Reduced amount of Dicer and Drosha have also been reported in mesenchymal stromal cells from human patients with myelodysplastic syndrome (MDS)[51], suggesting that global dysregulation of miRNAs may cause hematopoietic diseases like MDS. Thus, it appears that Drosha activity and proper production of miRNAs in HSCs are important for both embryonic and adult hematopoiesis. *Let-7* miRNAs have been implicated in the lineage commitment to natural killer T (NKT) cells and fetal B lymphocytes by targeting key transcription factors PLZF[52] and Arid3a[53], respectively. It has also been shown that Lin28b-dependent regulation of *let-7* controls the self-renewal potential of fetal and adult HSCs by modulating its target Hmga2[54]. The abundance of *Hmga2* mRNAs in HSPCs from cKO was only moderately increased unlike *Alox5* mRNAs (Supplementary Data 1).

Previous studies have implicated the roles of other arachidonic acid metabolites, such as prostaglandins (PGs) and epoxyeicosatrienoic acids (EETs) during embryonic hematopoiesis. PGE2, which is synthesized by enzymes COX1 and COX2, was identified as a mediator of HSC expansion in zebrafish and mouse[55]. 11,12-EET, which is synthesized by enzymes CYP2C and CYP2J, is known to promote a unique activator protein 1 (AP-1) and Runx1 transcription program to orchestrate cellular processes, such as migration, that ultimately facilitate HSPC engraftment in both zebrafish and mouse[56]. Our study adds a member of another class of arachidonic acid metabolites, LTB4, to the list of modulators of HSPC development, with one critical difference: unlike PGE2 and 11,12-EET, which are stimulatory to HSPC development, the effect of LTB4 at a high dose is inhibitory to HSPC development. Currently, no molecular mechanisms downstream of PGE2 or 11,12-EET in the context of HSPC development have been identified. The mechanism underlying the intricate network of different classes of arachidonic acid metabolites (LTB4, PGE2, and 11,12-EET) regulating HSPC biology, and especially the crosstalk of their downstream signaling pathways, needs to be elucidated in the future.

LTB4 has been implicated in a wide range of inflammatory conditions, including vascular disorders, such as atherosclerosis and pulmonary artery hypertension[39, 44]. LTB4 plays a binary role in inflammatory responses, partly as a potent chemoattractant of leukocytes and partly as a signaling molecule that binds G-protein coupled BLT1 and BLT2 receptors and triggers intracellular effects[44]. LTB4 is typically released from the cell into the extracellular milieu and acts non-cell autonomously, but it can also act cell-autonomously in the nucleus as a modulator of transcription[57]. Here we uncover the dose-dependent effect of the LTB4-BLT1 signaling pathway in HSPCs during developmental hematopoiesis. Our results demonstrate that HSPC clusters increase their adhesion to the vascular wall through induction of α4β1 Itg in response to LTB4. This is analogous to the inflammatory process by which leukocytes secreting LTB4 increase their adhesion to the vascular wall[39, 45, 58–61]. Therapies to block adhesion molecules, such as α4 Itg, are found to be effective for various inflammatory conditions, such as asthma, multiple sclerosis, and inflammatory bowel disease[48, 62, 63].

Previous studies demonstrate the essential role of Itgα4β1 in HSCs for homing into the FL, spleen, and bone marrow[64]. Indeed, we terminated the anti-Itgα4 antibody treatment at E11.5 to avoid a prolonged blockade of Itgα4 possibly disrupting liver homing of HSPCs. We speculate that the amount of α4β1 Itg on the surface of HSPCs is dynamic during maturation and likely being precisely controlled by multiple mechanisms, one of which is the Alox5-LTB4-BLT1 pathway. Besides the role as cell adhesion molecules, Itgs can also play a role in activating growth factor signaling, including transforming growth factor-β[65]. Thus, the regulation of α4β1 Itg is crucial not only for the interaction of HSPCs with the vascular microenvironment through cell adhesion, but may also play an important role in HSPC maturation through signaling. In conclusion, our study uncovers a previously unappreciated role of innate LTB4 signaling, and the mechanism by which miRNAs regulate *Alox5*, which in turn modulates LTB4 production, to tightly control Itg-mediated HSPC adhesion. This finely tuned regulation allows for HSPC maturation and eventual release from the vascular bed with subsequent migration to the FL.

## Methods

**Animal care and use**. All animal experiments were conducted in accordance with University of California at San Francisco Laboratory Animal Research Committee guidelines. *Cdh5-Cre*[19], *Drosha*[tm1Litt] floxed line[18], *Gt(ROSA)26Sor*[tm4(ACTB-tdTomato,-EGFP)Luo] line[20], and *Gata2*[tm1Sac] [36] have been previously described. All the mice were on the mixed background of 129 and C57BL/6J. Embryos were dated by the presence of vaginal plug in the female mouse as E0.5. Embryos both genders were used.

**Flow cytometry and cell sorting**. FL or AGM was dissected from embryos and mechanically dissociated by pipetting into single cell suspension in Hank's balanced salt solution containing 2% fetal bovine serum (FBS), 1% penicillin/streptomycin and buffered with 10 mM HEPES, pH7.2 (FACS buffer). Cells were stained with fluorescein conjugated antibody at 4 °C for 1 h, washed with DAPI containing FACS buffer and analyzed by FACS Verse (BD Biosciences) or sorted on a FACS Aria III (BD Biosciences) located at the UCSF FACS core. For flow analysis of peripheral blood, E13.5 embryos were harvested. Each embryo was bled in 500 μl FACS buffer. Whole peripheral blood was then stained with fluorescein-conjugated antibody at 4 °C for 1 h. After staining, red blood cells were lysed with RBC lysis buffer (Biolegend 420301) following manufacture's manual. The peripheral blood sample was then washed with DAPI (0.5 μg ml$^{-1}$) containing FACS buffer and analyzed by FACS Verse (BD Biosciences). Data were analyzed with FlowJo v10.0.7. IgG staining was used as a control for gating. Results shown in figures were normalized with average of Ctr in each litter.

**Cellularity analysis**. Cellularity of FL was analyzed according to previous study[26]. Briefly, E13.5 FL were dissociated into single-cell suspension in FACS buffer by pipetting and stained with DAPI at 4 °C for 30 min, followed by the analysis of total number of DAPI⁻negative (DAPI⁻) cells by FACS Verse (BD Biosciences). Total number of DAPI⁻ cells per FL relative to average of total number of DAPI⁻ cells from WT or control littermates' FLs are plotted as mean ± SEM.

**Antibodies**. The following antibodies directed against mouse antigens were used: Percp-CD45 (Biolegend 103130), FITC-CD31 (BD 553372), APC-CD117/Kit (BD

553356), PE-CD31 (Biolegend 102407), FITC-Ter119 (BD 561032), Percp-IgG2bk (Biolegend 400336), FITC-IgG2ak (BD 553929), APC-IgG2bk (BD 556924), PE-IgG (Biolegend 405307), PE-Annexin V (BD560930), DAPI (ThermoFisher D1306), PE-ItgαV (Biolegend 104106), and PE-Itgα4 (BD 557420). For flow analysis, antibodies were used at a concentration of $2\,\mu g\,ml^{-1}$.

**In vivo anti-Itgα4 antibody treatment**. Mice were treated with anti-α4 Integrin antibody (Millipore, CBL1304, clone PS/2) or control IgG-B (Santa Cruz, sc-2763). Antibodies ($2\,mg\,kg^{-1}$ per mouse) were given intraperitoneally at E9.5, E10.5 and E11.5. E12.0 AGM or E14.5 FL was harvested for subsequent analysis. To collect peripheral blood, E13.5 embryos were harvested and bled in FACS buffer, followed by antibody staining. Red blood cells in peripheral blood were then lysed with RBC lysis buffer (Biolegend 420301), followed by flow analysis.

**Cell culture and luciferase assay**. Mouse macrophage RAW264.7 cells (a gift from Dr. R. Swanson at UCSF) or HEK293 cells (ATCC) were cultured in 10%FBS in high-glucose DMEM with penicillin and streptomycin at 37 °C, 5% $CO_2$. Thirty-four nucleotide sequence partially complementary to *let-7* miRNA found in the 3′UTR of *Alox5* mRNA (5′-CAATAAAAAAGCTGGTCTACTACCTCCTCCA ACG-3′) was cloned at *SacI* and *NheI* site of the firefly luciferase miRNA sensor *pISO* (Addgene, plasmid #12178)[66] (pISO-AL5). pISO, pISO-AL5 (100 ng each), and renilla luciferase plasmid (1 ng) were transfected with 6 pmol *let-7e* or control (*miR-142-5p*) mimic (mirVana® miRNA mimic, Thermo Fisher Scientific) into one well of a 24-well plate of HEK293 cells. Total cell lysates were prepared 48 h after transfection, and subjected to the luciferase assay with Glomax 96 Microplate Luminometer (Promega) following manufacture's manual. The firefly luciferase activity was normalized by the renilla luciferase activity to normalize transfection efficiency. Results are shown as a ratio of pISO-AL5/pISO after normalization by renilla luciferase activity.

**Next generation sequencing**. HSPC cluster cells (CD31+CD45−Kit+) and ECs (CD31+CD45−Kit−) cells were sorted from five cKO, eight WT, and five Het embryos. Cells from embryos with the same genotype were pooled for RNA preparation with RNeasy Plus micro kit (Qiagen 74034). The quality of RNAs was evaluated with 2100 Bioanalyzer Instrument (Agilent Technologies). RNAs samples with RNA integrity number > 8.0 were shipped to Beijin Genome Institute for a library preparation and sequencing (Illumina HiSeq 2500).

**Quantitative RT-PCR analysis**. For qRT-PCR analysis, total RNA was prepared in the same way as next generation RNA-sequencing. 5 ng RNA from HSPC cluster cells and 50 ng RNA from ECs were amplified and reverse transcribed with a Nugen ovation picoSL WTA system V2 (Nugen 3312-24). Samples were then 1:100 diluted for qRT-PCR analysis. For qPCR analysis of E10.5 $Alox5^{+/-}$ and $Alox5^{+/+}$ embryos, E10.5 whole embryos were lysed in TRIzol (Ambion, 15596018) for RNA extraction and reverse transcriptase reaction was performed using iScript™ cDNA Synthesis Kit (BIO-RAD, #1708891). qRT-PCR reactions were performed in triplicates using iQ SYBR Green supermix (Bio-RAD 1708882)[67]. Primers for qRT-PCR are listed in Supplementary Table 5.

**Genotyping of mouse**. Genomic DNAs were isolated from tail tips of postnatal day 12 mice or conceptus yolk sacs and genotyped using regular PCR. Primers for genotyping are listed in Supplementary Table 5.

**Histology immunofluorescence staining and microscopy**. E13.5 embryos were harvested by cesarean section, rinsed in PBS(-), PH 7.4, fixed in 4% paraformaldehyde overnight at 4 °C, then embedded in paraffin, sectioned at a thickness of 7 μm, and stained with hematoxylin and eosin. Sections were imaged with Nikon E1000 microscope. For IF staining, E9.5 and E10.5 embryos were fixed in 4% paraformaldehyde solution overnight, dehydrated in 30% sucrose in PBS(-) and then embedded in Tissue-Tek OCT Compound (Sakura Finetec, 4583). Cryosections (40 μm) were done with Leica CM1850V. Slides were dried for 2 h at room temperature and stored at −80 °C before IF staining. Cryosections were warmed to room temperature, washed in PBS, penetrated with 0.5% Triton-X-100 in PBS(-) for 10 min and then blocked using 3% BSA 0.05%/Triton-X-100 in PBS(-) for 1 h at room temperature. Slides were incubated with primary antibodies at 4 °C overnight, followed by PBS wash and secondary antibody incubation for 1–2 h at room temperature. Images were taken with Leica SPE Confocal Microscope and complied with ImageJ software. All the experiments were performed with three biological replicates. For whole-mount IF staining, AGM of E10.5 embryos were dissected and fixed in 4% paraformaldehyde solution overnight, dehydrated in series of methanol/PBS, and stored at −20 °C before the experiment. AGMs were rehydrated through a series of methanol (25, 50, and 75%)/PBS(-) solution, blocked using 3% milk/0.1%Tween in PBS(-) for 2 h at room temperature, then incubated with primary antibody for 48 h at 4 °C. AGMs were then washed with 3% milk/ 0.1%Tween in PBS(-) for 5 times, 1 h each time, and the last wash was overnight at 4 °C. Secondary antibody was incubated for 48 h at 4 °C. AGMs were then dehydrated through a series of methanol/PBS, cleared and mounted with BABB solution (benzyl alcohol: benzyl benzoate). Images were taken with Leica SPE Confocal

Microscope and complied with ImageJ software. The following antibodies directed against mouse antigens were used: CD31 (BD pharmaceuticals, 550274), Runx1 (Abcam, ab92336), Sox17 (Abcam, ab155402), anti-rat IgG 594 (ThermoFisher, A-21209), and anti-mouse IgG 488 (ThermoFisher, A-21202). All the experiments were performed with three biological replicates. Primary antibodies were used at a concentration of 500 ng ml$^{-1}$. Secondary antibodies were used at a concentration of $5\,\mu g\,ml^{-1}$.

**Scanning electron microscopy**. E10.5 embryos were harvested by cesarean section, fixed in 2% paraformaldehyde solution at 4 °C overnight, washed in PBS(-), embedded in 4% low melting point agarose[68]. The embryos were then sectioned sagittally on a vibratome (Leica VT 100P) at 100–300 μm, peeled out of the agarose and refixed in 0.1 M sodium cacodylate, followed by dehydration in a series of ethanol washes (30, 50, 70, 90, and 100%). Samples can be stored at 4 °C for up to 1 week in 100% ethanol. Samples were dried in a critical point dryer, coated with 8 nm of iridium labeling prior to image acquisition on a Zeiss Ultra55 FE-scanning electron microscope. Imaging was performed with five biological replicates for every genotype. More than 20 images were taken for each embryo and 10 images were used for quantitative analysis.

**In vivo treatment of mice with antagonists**. 5 mg kg$^{-1}$ zileuton (Cayman #10006967) or 0.5 mg kg$^{-1}$ U75302 (Cayman #70705) was injected intraperitoneally in pregnant female mice at E9.5 and E10.5. Embryos were harvested on E11.5. AGM and FL were dissected and subjected to CFU assay.

**In vivo treatment of mice with LTB4**. LTB4 (Cayman #20110) was diluted with normal saline. Because LTB4 is dissolved in ethanol, the same volume ethanol was diluted with normal saline and used as control reagent. 750 μM LTB4 or control reagent was injected intraperitoneally into pregnant female mice daily from E8.5-E10.5. At E11.5, embryos were dissected, FL and AGMs were taken and subjected to CFU assay.

**Enzyme-Linked Immunosorbent Assay**. HSPC (CD31+CD45−Kit+) and EC (CD31+CD45+Kit+) populations from E11.5 Ctr or cKO AGM were sorted by FACS Aria III (BD Biosciences) into 7% BSA in FACS buffer. Cells were precipitated by centrifugation at 2500 RPM, 5 min, 4 °C. Cells were lysed in PBS(-) by freeze-and-thaw for three times using liquid nitrogen and 37 °C water bath. Cells were then stored at −80 °C prior to ELISA. ECs from two embryos of the same genotype and early HSPCs from five embryos of the same genotype were pooled and the level of LTB4 was measured using LTB4 Parameter Assay kit (R&D KGE006B) according to manufacturer's instructions.

**In vivo cell proliferation analysis**. The rate of cell proliferation in vivo was measured by 5′-ethynyl-2′-deoxyuridine (EdU) incorporation into DNA followed by image analysis using the Click-iT™EdU Alexa Fluor Imaging Kit (Invitrogen/ Molecular Probes C10337) or flow analysis with Click-iT EdU Alexa Flour 488 Flow Cytometry Assay Kit (Invitrogen/Molecular Probes C10425). For imaging analysis, E10.5 pregnant female mice were injected with EdU at a dose of 50 mg kg$^{-1}$ 2 h prior to euthanization and embryos were harvested. Embryos were fixed in 4% paraformaldehyde solution at 4 °C overnight, and cryosectioned at a thickness of 50μm. Sections were then washed with PBS(-) and permeabilized with 0.5% Triton X-100 in PBS(-) for 20 min, incubated with a Click-iT™ reaction cocktail for 1 h for EdU staining, followed by immunostaining with CD31 antibody and imaging with Leica SPE confocal microscope. The image EdU analysis was performed with biological duplicates and technical triplicates. For flow analysis, E10.5 pregnant female mice were injected with EdU at a dose of 50 mg kg$^{-1}$ 1 h prior to be euthanized and embryos were harvested in FACS buffer. AGM was pipetted into single-cell suspension, washed with PBS(-) and stained with Fixable Viability Dye eFluor 450 (eBioscience, 65-0863-14). Cells were then fixed, permeabilized, stained with Click-iT EdU reaction cocktail, followed by wash and cell surface marker staining, and flow analysis. Ten control embryos and five cKO embryos from two litters were subjected to EdU flow analysis.

**OP9-DL1 co-culture**. Murine bone marrow-derived stromal cell line OP9-DL1[69] were cultured in 20% FBS in αMEM supplemented with 5 ng ml$^{-1}$ recombinant human Flt-3L (R&D Systems, 308-FK), 1 ng ml$^{-1}$ recombinant murine IL-7 (Peprotech, 217-17), and penicillin-streptomycin1. CD31+ cells (1,500 cells) were sorted onto OP9-DL1 monolayers and cultured for 7 days. The HSPC clusters were then mechanically scraped and flushed out, followed by for CFU assay.

**Explant culture of AGM and colony formation unit assay**. AGMs of E10.5 embryos were dissected and cultured at 37 °C for 20 h on 40 μm filters at an air-liquid interface in myelocult medium (Stem Cell Technologies M5300) supplemented with 10 μM hydrocortisone. AGMs from E9.5 embryos were cultured under the same conditions with additional supplements, such as 100 ng ml$^{-1}$ SCF (PeproTech), 10 ng ml$^{-1}$ murine Oncostatin M (R&D) and 1 ng ml$^{-1}$ basic FGF (R&D). Then, 50 μM zileuton (Cayman 10006967), 2 μM U75302 (Cayman #70705) or 1–100 nM LTB4 (Cayman #20110) were added as indicated. Zileuton was

dissovled in 50% DMSO/50% PBS(-). After the culture, AGMs were then pipetted into single-cell suspension and seeded into methocult medium (Stem Cell Technologies #M3434). The number of CFU-GM, BFU-E and CFU-GEMM were counted after 7–10 days culture. For CFU assay, E13.5 FL or E10.5 AGM was harvested and dissociated into single-cell suspension. Then, 0.13% of E13.5 FL suspension, 50% of E10.5 AGM or 10% of E11.5 AGM or FL suspension was seeded into methocult medium (Stem Cell Technologies M3434) and cultured for 7–10 days prior to counting the number of CFU-GM, BFU-E and CFU-GEMM colonies. For transfection of let-7e mimic or control mimic (miR-142-5p), 18 nM miRNA mimics were transfected into cultured E10.5 AGM with Lipofectamine RNAiMAX (Thermo Fisher Scientific) following manufacturer's instructions.

**TUNEL assay.** Terminal deoxynucleotidyl transferase (TdT) dUTP Nick-End Labeling (TUNEL) assay was performed with E10.5 AGM cryosection according to manufacturer's instructions (In Situ Cell Death Detection Kit, Fluorescein, Roche, 11684795910). Cryosections were washed with PBS(-), permeabilized with 0.2% Triton X-100 in PBS for 10 min at room temperature, blocked using 3%BSA 0.02% Triton X-100 in PBS for 1 h at room temperature, and incubated with rat anti-CD31antibody(BD pharmaceuticals, 550274) overnight at 4 °C. Anti-rat Alexa Fluor 594 secondary antibodies (Molecular Probes) were added to the TUNEL reaction mix, which was prepared by diluting 1 part of enzyme solution in 9 parts of label solution from the kit (Roche, 11684795910). The sections were incubated with the secondary antibody/TUNEL reaction mix for 1 h at 37 °C, washed in PBS three times, incubated with DAPI for 5 min, and mounted in Vectashield (Vector Laboratories) for microscopy. The experiment was performed for five times with biological triplicates of each genotype.

**Statistical analysis.** Graphs were generated with GraphPad PRISM software. Statistical significance was calculated in R version 3.2.3 by Student's $t$ test. All data sets were considered paired. The null hypothesis of the medians/means being equal was rejected at $\alpha = 0.05$ and $p$ values were generated by unpaired Student's $t$ test and presented in figures. The sample size is presented in the figure legend. For animal analysis, at least three individual animals were used in each experiment and all experiments were completed in gender- and genotype-blind manner. No animals were excluded. The investigators were blinded during experiments, because genotyping was done after experiments. Experiments were not randomized.

**Data availability.** The authors declare that all data supporting the findings of this study are available within the article and its supplementary information files or from the corresponding author upon reasonable request. Sequencing data have been deposited in the NCBI-SRA database under accession codes (SRA 3644408-3644422).

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

## Acknowledgements

We thank Drs. D. Sheppard, W. DeGrado, N. Reed, J. Hyunil at UCSF for providing the integrin inhibitor and assisting with the delivery to animals. We also thank Dr. R. Swanson at UCSF for providing the RAW264.7 cells. We thank Dr. Juan Carlos Zúñiga-Pflücker at Sunnybrook Health Sciences Centre for the OP9-DL1 cell lines. We thank all members of the Hata lab for critical discussion and reading of the manuscript. This work was supported by grants from the NHLBI/NIH (HL093154 and HL108317) and Fondation LeDucq to A.H. and Burroughs Welcome Fund Career Award for Medical Scientists (1008408.01) and the NIH Office of the Director's Innovator Award (1DP2HL117743-01) to A.C.Z.

## Author contributions

X.J. and J.S.H. designed and performed the experiments, and interpreted the data. F.L.B performed the scanning electron microscopy imaging. J.Le helped with the CFU assay and mouse breeding. P.G. and J.Lo performed all the genotyping. C.O.L. and G.L. interpreted the data, and edited the manuscript. A.C.Z., A.H., and X.J. conceived the project, designed the experiments, interpreted the data, and wrote the manuscript. A.H. directed the overall project. All authors contributed to reading and editing the manuscript.

## Additional information

**Competing interests:** The authors declare no competing financial interests.

