## [Peer Review File · Nature Communications]

This manuscript has been previously reviewed at another journal that is not operating a transparent peer review scheme. This document only contains reviewer comments and rebuttal letters for versions considered at Nature Communications.

Reviewers' Comments:

Reviewer #1:

Remarks to the Author:

In the revised manuscript the authors have sufficiently addressed my comments in the previous review with Nature Cell Biology. Hence, I recommend the publication of this manuscript in Nature Communications.

Reviewer #3:

Remarks to the Author:

The authors have answered most of my comments well; however, there are just a few issues that need clearing up and a few corrections that need to be made, as detailed below.

Major comments

1. I agree with reviewer 2 that the phenotype of the VEC-Cre-specific Droscha knockout is likely to be a composite of multiple events mediated by several different miRNAs. While the authors have convincing evidence that one of these events involves the let-7 family of miRNAs via LTB4 and Itga4, it should be made clearer, especially in the abstract (e.g. line 9; and also line 18 on page 4), that the overall phenotype is only partly caused by the depletion of the let-7 family.
2. Some questions remain as to how exactly HSPC generation is affected. The authors claim that HSPCs emerge normally (e.g. line 7 of the abstract and line 18 on page 4), but it is clear from the CFU-C assays and OP9 co-cultures that despite these cells appearing in clusters and having a cell surface phenotype resembling HSPCs (i.e. CD41 and ckit expression), they clearly have no hematopoietic potential. If it were all due to the failure of Itga4 downregulation and release from the clusters, then dissociating them mechanically and plating them in methylcellulose should reveal their hematopoietic identity. However, the authors' own results show that there is clearly a failure to complete hematopoietic transition and the failure to dissociate from the clusters may be a consequence of that. Therefore, the sentence in line 7 of the abstract (and wherever else this statement is made) needs to be altered to reflect that, e.g. "Despite phenotypic HSPCs emerging normally..."
3. I am struggling to identify the structures that are shown in the immunohistochemistry images in Fig. 2B. Are those the somites at the bottom with the intersomitic vessels highlighted by CD31 staining? Are the clusters on the dorsal side then?
4. Fig. 2A and 3A are actually the same thing. They show FACS plots with the same 3 markers and the same tissue from the same time point, just with slight variations on how the markers are combined. Are they actually the same data? It seems a bit superfluous.
5. The authors claim on page 10, line 6 that by E11.5 the CD45+ hematopoietic populations were further reduced, but the numbers at E10.5 and E11.5 are exactly the same: cKO: E10.5 – 0.9%, E11.5 – 0.94%; Ctr: E10.5 – 2.0%, E11.5 – 2.11%.
6. Is there really an increase in the absolute numbers of CD31+ckit+CD45- cells in the E12 AGM? The authors only show the percentages in Fig. 4C, which might be skewed due to the absence of CD45+ cells.
7. The authors were largely able to address my concern regarding the elevated levels of Gata2 contributing to the haematopoietic phenotype; however, I was a bit disappointed that they did not measure Gata2 expression levels in their cKO;Gata2-Het compound mice by qPCR as they had done for the Alox5 compound mice in Fig. S6F. If the levels of Gata2 were actually substantially reduced, then this could by itself cause a hematopoietic defect (Gata2+/- embryos have a hematopoietic defect as shown by KW Ling et al J Exp Med 2004).
8. I am quite concerned about how the levels of LTB4, as determined by ELISA, have changed between this version of the manuscript and the last. In the first version, the authors gave a value of intracellular LTB4 concentration in cKO HSPCs of 3.5pM/10⁶ cells (former manuscript Fig. 6B and page 14, line 12), whereas in the current version it is 52pM/1000 cells for the same population (Fig. 6B; page 15, line 7). That is a huge difference and cannot be explained by a different

normalisation method as the authors claim in their response to my comment (point 6). It somewhat casts a doubt on the accuracy of the measurement and makes one wonder which one is actually correct. Also, the current figure has no longer any error bars. Importantly, the authors claim that LTB4 levels in control HSPC cells are below detection (previous manuscript page 14, line 13) and they make a big point of that in their response to my major comment 5. Yet, in the current version of the manuscript they give a value of 23pM/1000 cells for the wild-type/Ctr HSPCs on page 15, line 7 and in Fig. 6B. How is that possible? It also needs to be clarified in the figure legends or the materials and methods that the concentration is per 1000 cells.

9. For each figure showing colony-forming data, the units on the y axes need to be clarified, i.e. number of colonies/number of cells plated, especially since there are big differences between experiments, e.g. E11.5 Ctr AGM cells gave more than 300 colonies in Fig. 5D, while they only give just over 40 colonies in Fig. 6E.

Minor comments

1. The frequency of Droscha fl/fl; Cdh5-cre+ pups at E15.5 in Supplementary Table 2 needs correcting. It is not 0.03%, but 3%.
2. The y axis label in the top left FACS plot in Fig. 2A needs to be corrected to CD31.
3. P.9, line 5: after the rearrangement of the figures for the revisions, the qPCR plot is no longer at the bottom of Fig. 2D.
4. The cKO and Ctr values are swapped in line 8 on page 10: it should be 2.90% for cKO and 6.64% for Ctr (Fig. 3Bd).
5. I assume that the scatter plot in Fig. 4Cd should have "%DAPI-" as the y axis label.
6. Page 13, line 6: The expression of Alox5 and Alox5-AP is now shown in Fig. 5B (not 5A).
7. Page 13, line 7: The expression of Pla2g6 is not shown in Fig. 5.
8. Page 14, line 10: the authors conclude at this point in their manuscript that a tight regulation of LTB4 is required. However, this comes a little bit out of the blue, since it is only determined in the following section which leukotriene is involved. LTB4 should therefore be changed to LT.
9. Page 16, line 6: the colony numbers following U75302 treatment is shown in Fig. S6H (not S6G).
10. Page 17, line 16: the colony counts are no longer shown at the bottom of Fig. 7C.

Reviewer #3:

Major comments

1. I agree with reviewer 2 that the phenotype of the VEC-Cre-specific Droscha knockout is likely to be a composite of multiple events mediated by several different miRNAs. While the authors have convincing evidence that one of these events involves the let-7 family of miRNAs via LTB4 and Itga4, it should be made clearer, especially in the abstract (e.g. line 9; and also line 18 on page 4), that the overall phenotype is only partly caused by the depletion of the let-7 family.

The text was edited accordingly.

2. Some questions remain as to how exactly HSPC generation is affected. The authors claim that HSPCs emerge normally (e.g. line 7 of the abstract and line 18 on page 4), but it is clear from the CFU-C assays and OP9 co-cultures that despite these cells appearing in clusters and having a cell surface phenotype resembling HSPCs (i.e. CD41 and ckit expression), they clearly have no hematopoietic potential. If it were all due to the failure of Itga4 downregulation and release from the clusters, then dissociating them mechanically and plating them in methylcellulose should reveal their hematopoietic identity. However, the authors' own results show that there is clearly a failure to complete hematopoietic transition and the failure to dissociate from the clusters may be a consequence of that. Therefore, the sentence in line 7 of the abstract (and wherever else this statement is made) needs to be altered to reflect that, e.g. "Despite phenotypic HSPCs emerging normally..."

The text was edited accordingly.

3. I am struggling to identify the structures that are shown in the immunohistochemistry images in Fig. 2B. Are those the somites at the bottom with the intersomitic vessels highlighted by CD31 staining? Are the clusters on the dorsal side then?

The clusters are found on the ventral side of the DA. The small vessel highlighted by CD31 is superior mesenteric artery located on the ventral side. To better orient the image, the dorsoventral axis was indicated in Fig. 2B.

4. Fig. 2A and 3A are actually the same thing. They show FACS plots with the same 3 markers and the same tissue from the same time point, just with slight variations on how the markers are combined. Are they actually the same data? It seems a bit superfluous.

According to this comment, we deleted the original Fig. 3A.

5. The authors claim on page 10, line 6 that by E11.5 the CD45+ hematopoietic populations were further reduced, but the numbers at E10.5 and E11.5 are exactly the

same: cKO: E10.5 – 0.9%, E11.5 – 0.94%; Ctr: E10.5 – 2.0%, E11.5 – 2.11%.

Because the degree of reduction of CD45+ populations is similar between E10.5 and E11.5, and following comment #4, we have deleted Fig. 3A. We also edited the sentence to “CD45+ hematopoietic populations were reduced in cKO AGMs at E11.5”.

6. Is there really an increase in the absolute numbers of CD31+ckit+CD45- cells in the E12 AGM? The authors only show the percentages in Fig. 4C, which might be skewed due to the absence of CD45+ cells.

There is an actual increase in the cell number of CD31+kit+CD45- populations. The plot with “actual cell number” on the Y-axis is shown here (right).

7. The authors were largely able to address my concern regarding the elevated levels of Gata2 contributing to the haematopoietic phenotype; however, I was a bit disappointed that they did not measure Gata2 expression levels in their cKO;Gata2-Het compound mice by qPCR as they had done for the Alox5 compound mice in Fig. S6F. If the levels of Gata2 were actually substantially reduced, then this could by itself cause a hematopoietic defect (Gata2^{+/-} embryos have a hematopoietic defect as shown by KW Ling et al J Exp Med 2004).

First, we clarify that the original Fig. S6F (revised Fig. S7F), we show the expression of Alox5 and ITGA4 mRNA in Alox5 WT or heterozygous null mice (not a compound mouse as the reviewer commented above). In the original Fig. S5D (now Fig. S6), we show that Gata2 Hets (Drosha Ctr) reveal no hematopoietic defects, unlike the report by KW Ling et al J Exp Med 2004. This discrepancy is possibly due to the fact that Ling et al studied “conventional” GATA2(+/-) mice, while we used a conditional mouse with a heterozygous deletion of GATA2 in the endothelium (Gata2^{flox/+}:Cdh5-Cre mouse.) Because our Gata2 Het (Drosha Ctr) mice show no hematopoietic phenotype, the hematopoietic defects in Drosha cKO:GATA2 Het compound mice is due to conditional knock-out of *Drosha*. Furthermore, GATA2 heterozygosity was unable to rescue the Drosha cKO phenotype, which is the main point of this experiment.

8. I am quite concerned about how the levels of LTB4, as determined by ELISA, have changed between this version of the manuscript and the last. In the first version, the authors gave a value of intracellular LTB4 concentration in cKO HSPCs of 3.5pM/10ex6 cells (former manuscript Fig. 6B and page 14, line 12), whereas in the current version it is 52pM/1000 cells for the same population (Fig. 6B; page 15, line 7). That is a huge difference and cannot be explained by a different normalisation method as the authors claim in their response to my comment (point 6). It somewhat casts a doubt on the accuracy of the measurement and makes one wonder which one is actually correct. Also, the current figure has no longer any error bars. Importantly, the authors claim that LTB4 levels in control HSPC cells are below detection (previous manuscript page 14,

line 13) and they make a big point of that in their response to my major comment 5. Yet, in the current version of the manuscript they give a value of 23pM/1000 cells for the wild-type/Ctr HSPCs on page 15, line 7 and in Fig. 6B. How is that possible? It also needs to be clarified in the figure legends or the materials and methods that the concentration is per 1000 cells.

We would like to clarify that in the original manuscript, we calculated the “amount” of LTB4 as 3.5 **pmol** (not **pM**) per 1 million cells as shown here (right). In the revised manuscript, we calculated the “concentration” of LTB4 as 52 **pM** per 1,000 cells in 50 μ l lysates. Therefore, there is no significant discrepancy.

9. For each figure showing colony-forming data, the units on the y axes need to be clarified, i.e. number of colonies/number of cells plated, especially since there are big differences between experiments, e.g. E11.5 Ctr AGM cells gave more than 300 colonies in Fig. 5D, while they only give just over 40 colonies in Fig. 6E.

As the reviewer indicates, we noted the insufficient information on the number of cells or proportion of cells subjected to the CFU assay. We now provide this information in the figure legend of all CFU assay results. Regarding the discrepancy of colony numbers in Fig. 5D versus Fig. 6E: in Fig. 5D, all cells from AGM or FL were subjected to CFU assay, while in Fig. 6E, only 20% of AGM and 10% FL cells were subjected to CFU assay. This explains why approximately 1/5 and 1/10 of colony numbers in Fig. 5D were obtained in Fig. 6E.

Minor comments

1. The frequency of Droscha fl/fl; Cdh5-cre+ pups at E15.5 in Supplementary Table 2 needs correcting. It is not 0.03%, but 3%.

We corrected the mistake.

2. The y axis label in the top left FACS plot in Fig. 2A needs to be corrected to CD31.

We corrected the mistake.

3. P.9, line 5: after the rearrangement of the figures for the revisions, the qPCR plot is no longer at the bottom of Fig. 2D.

We corrected the mistake.

4. The cKO and Ctr values are swapped in line 8 on page 10: it should be 2.90% for cKO and 6.64% for Ctr (Fig. 3Bd).

We corrected the mistake.

5. I assume that the scatter plot in Fig. 4Cd should have “%DAPI-“ as the y axis label. The Y-axis should read “CD31” as it is.

6. Page 13, line 6: The expression of Alox5 and Alox5-AP is now shown in Fig. 5B (not 5A).

We corrected the mistake.

7. Page 13, line 7: The expression of Pla2g6 is not shown in Fig. 5.

We corrected the mistake.

8. Page 14, line 10: the authors conclude at this point in their manuscript that a tight regulation of LTB4 is required. However, this comes a little bit out of the blue, since it is only determined in the following section which leukotriene is involved. LTB4 should therefore be changed to LT.

We corrected the mistake.

9. Page 16, line 6: the colony numbers following U75302 treatment is shown in Fig. S6H (not S6G).

We corrected the mistake.

10. Page 17, line 16: the colony counts are no longer shown at the bottom of Fig. 7C.

We corrected the mistake.

Reviewers' Comments:

Reviewer #3:

Remarks to the Author:

The authors' response to my queries is sufficient and I have no further comments.